# LRQ: Optimizing Post-Training Quantization for Large Language Models by Learning Low-Rank Weight-Scaling Matrices

## Abstract

With the commercialization of large language models (LLMs), weight-activation quantization has emerged to compress and accelerate LLMs, achieving high throughput while reducing inference costs. However, existing post-training quantization (PTQ) techniques for quantizing both weights and activations of LLMs still suffer from non-negligible performance drops, especially on massive multitask language understanding. To address this issue, we propose Low-Rank Quantization (LRQ) - a simple yet effective post-training weight quantization method for LLMs that reconstructs the outputs of an intermediate Transformer block by leveraging low-rank weight-scaling matrices, replacing the conventional full weight-scaling matrices that entail as many learnable scales as their associated weights. Thanks to parameter sharing via low-rank structure, LRQ only need to learn significantly fewer parameters while enabling the individual scaling of weights, thus boosting the generalization capability of quantized LLMs. Through extensive experiments, we demonstrate the superiority of LRQ to prior LLM INT8 PTQ works. Remarkably, we for the first time conduct experiments on 4-bit weight and 8-bit activation quantization for LLMs with minimal accuracy loss among LLM PTQ studies.

## 1 Introduction

As ChatGPT and GPT-4 (OpenAI, 2023) have showcased unprecedented capabilities across various domains such as common sense reasoning, mathematical problem-solving, and coding proficiency, there has been an exponential surge in interest surrounding the development of Large Language Models (LLMs). This surge in interest has culminated in the recent release of cutting-edge LLMs like Llama (Touvron et al., 2023a), PaLM 2 (Google et al., 2023), and Llama 2 (Touvron et al., 2023b). Accordingly, serving LLMs has rapidly emerged as a significant concern in both academia and industry. This stems from the substantial memory footprint and considerable computational cost incurred when operating these language models with tens or hundreds of millions of parameters in FP16 format. Therefore, extensive efforts (Frantar et al., 2023; Liu et al., 2023b) such as quantization or pruning are underway to compress LLMs and provide efficient deployment. In particular, quantization has garnered considerable interest among LLM engineers and researchers because quantization aids in not just model compression but also inference acceleration.

LLM quantization techniques fall into two primary categories: weight-only quantization and weight-activation quantization. Weight-only quantization concentrates on enhancing memory-bound operations like matrix-vector multiplication by quantizing weights of LLMs into low-bit integers (e.g., 2-4 bits). With activations kept in FP16, weight-only quantization exhibits marginal accuracy degradation but is only effective in accelerating text generation inference for small batch sizes (e.g., a single batch). In contrast, weight-activation quantization aims to expedite computationally intensive operations, such as matrix-matrix multiplication, typically by quantizing both weights and activations of LLMs into 8-bit integers and employing INT8 GEMM kernels. This comprehensive quantization approach enables LLM serving for large batch sizes, thus enhancing LLM throughput and expediting LLM inference through integer matrix multiplication. Yet, it comes with the trade-off of potential non-negligible accuracy drop. While each methodology boasts its own set of strengths and weaknesses, we focus on weight-activation quantization on the grounds that achieving high-throughput LLM inference is indispensable for handling a substantial volume of user requests in real time.

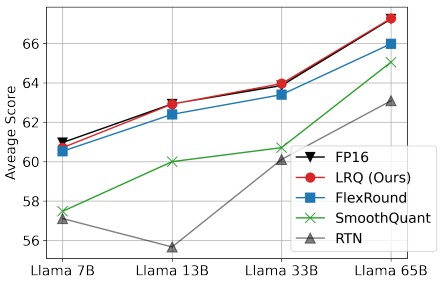 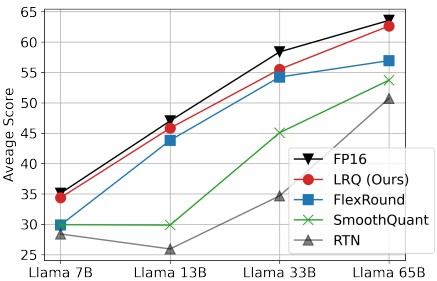

(a) Common Sense Reasoning tasks  (b) Massive Multitask Language Understanding

Figure 1: (a) Zero-shot performance and (b) five-shot performance of Llama with 8-bit per-channel asymmetric weight quantization and 8-bit *per-tensor* asymmetric *static* activation quantization.

Recent studies (Dettmers et al., 2022; Yao et al., 2022; Xiao et al., 2022; Lee et al., 2023; Liu et al., 2023a) have attempted to quantize both weights and activations of LLMs. Among these works, only SmoothQuant (Xiao et al., 2022) and FlexRound (Lee et al., 2023) demonstrated the potential for a hardware-efficient per-tensor static activation quantization scheme that can reduce the inference latency and memory usage by up to two-thirds and half respectively compared to FP16 baselines as thoroughly elucidated in Xiao et al. (2022). Given the compelling advantages of this scheme, we also stick mainly to per-tensor static activation quantization, with a primary focus on preventing non-negligible performance degradation, one of its key drawbacks, from occurring.

Despite promising results that SmoothQuant and FlexRound yielded, they still possess inherent limitations on enhancing model accuracy when using per-tensor static activation quantization. Although SmoothQuant is a potent technique for alleviating the difficulty of quantizing activation outliers, it uniformly divides activations in each channel and multiplies the weights in the corresponding input channel by some constant. Since such an uniform per-channel smoothing transformation can only scale the weights collectively per channel, not individually, SmoothQuant may lead to non-negligible accuracy loss after quantization for certain models as seen in Figure 1. On the other hand, as FlexRound learns a separate scale for each weight and thus enables flexible weight quantization based on individual characteristics of each weight, FlexRound can show marginal zero-shot accuracy drop on common sense reasoning tasks in Figure 1(a). However, as depicted in Figure 1(b), FlexRound falls short in performing well on massive multitask language understanding (MMLU), which necessitates problem-solving skills, specialized knowledge, as well as basic knowledge across diverse subjects. We empirically confirm that this phenomenon is because FlexRound has to learn too many scales relative to limited calibration samples due to the assignment of an independent scale to every weight.

To improve generalization performance on such a challenging benchmark as well, we propose a new post-training weight quantization approach, "Low-Rank Quantization (LRQ)". LRQ is designed to minimize the mean squared error between the outputs of an intermediate FP16 Transformer block and those of its quantized counterpart with respect to low-rank weight-scaling matrices instead of full weight-scaling matrices that involve as many scales as their associated weights. Through the use of such low-rank matrices, we can reduce the number of learnable parameters effectively while maintaining the concept of scaling weights individually by sharing learnable parameters via low-rank structure. As a result, LRQ can attain comparable accuracy to FP16 baselines on both common sense reasoning tasks and MMLU for all the Llama models as indicated in Figure 1.

Our main contribution is threefold:

- We propose a new post-training weight quantization method coined LRQ that leverages low-rank weight-scaling matrices for intermediate Transformer block output reconstruction, which improves the generalization performance of quantized LLMs as shown in Figure 1.

- We provide empirical insights into the significance of reducing the number of learnable parameters and how the utilization of low-rank matrices to effectively decrease learnable parameters impacts the generalization ability of quantized LLMs.

- We validate the effectiveness of LRQ for Llama and Llama 2 using per-tensor static activation quantization and for the first time perform experiments on 4-bit weight and 8-bit per-token activation quantization for LLMs with minimal accuracy loss among LLM PTQ approaches.

## 2 METHOD

In this section, we outline the post-training quantization (PTQ) background that our method, LRQ is founded on, figure out the problem arising when quantizing LLMs, and formulate LRQ. Finally, we deepen an empirical understanding of how LRQ can improve generalization in quantized LLMs.

### 2.1 BACKGROUND

**Block-wise Reconstruction**   First of all, our method is based on block-wise reconstruction, which originates from BRECQ (Li et al., 2021) for the purpose of taking into account the intra-block dependency and has been widely used in QDrop (Wei et al., 2022), FlexRound (Lee et al., 2023), and AQuant (Li et al., 2023) due to its efficacy to yield less generalization error than layer-wise reconstruction. As we concentrate on weight-activation quantization of LLMs that are generally Transformer-based models, the block-wise reconstruction process is applied to every Transformer block in the order of arrangement. To be more concrete, with a small set of calibration data, the objective of block-wise reconstruction is to find quantized weights $\widehat{W}$ by minimizing the block reconstruction error $\|WX - \widehat{W}\widetilde{X}\|_2^2$ where $W$ and $X$ are the weights and inputs of a FP16 Transformer block while $\widetilde{X}$ is the inputs of its quantized counterpart (i.e., the outputs of its immediately preceding Transformer block with all its previous Transformer blocks quantized).

**FlexRound**   Among PTQ studies that take advantage of block-wise reconstruction, FlexRound shows the state-of-the-art performance for a wide variety of models ranging from computer vision models to large language models including Llama. In FlexRound, the formulation of $\widehat{W}$ is written as

$$\widehat{W} = s_1 \left\lfloor \frac{W}{s_1 \odot \exp(S_2)} \right\rceil, \tag{1}$$

where $s_1$ is a quantization step size, $S_2$ is a weight-scaling matrix whose shape is exactly the same as that of $W$, $\lfloor \cdot \rceil$ and $\exp(\cdot)$ indicate the rounding and exponential function, and $\odot$ and $/$ represent element-wise multiplication and division. Depending on the type of $W$, some supplementary vectors are added to $S_2$, but we exclude these additional vectors to keep the expression uncluttered. At the beginning of learning, $S_2$ is set to a zero matrix and $s_1$ is initialized to $\arg\min_{s_1} \|W - \widehat{W}\|_2^2$ to start learning from rounding-to-nearest (RTN). Then, both $s_1$ and $S_2$ are learned to minimize the block reconstruction error $\|WX - \widehat{W}\widetilde{X}\|_2^2$ with a small amount of calibration data as explained above. As seen in Figure 1, even though the accuracy of FlexRound on common sense reasoning benchmarks nearly matches that of FP16 baselines, quantized LLMs via FlexRound might exhibit reduced scores on challenging tasks including massive multitask language understanding (MMLU).

### 2.2 MOTIVATION

We hypothesize that the failure to generalize well on challenging benchmarks like MMLU arises from the necessity of learning an individual scale for every weight with limited calibration samples. Now that $S_2$ has as many learnable parameters as the size of $W$ in Eq. 1, FlexRound's objective to achieve flexible weight quantization through the assignment of an independent scale to each weight may be deemed excessive when applied to LLM quantization. For instance, for Llama 7B, the smallest model in Llama, FlexRound has to learn more than 200 million scales with only just a few hundred or thousand calibration samples. FlexRound may be therefore prone to overfitting when quantizing LLMs. To resolve this issue, there might be two solutions: (i) increasing calibration samples, and (ii) decreasing learnable parameters. In the former case, as shown in Figure 2, the accuracy of FlexRound on MMLU does increase

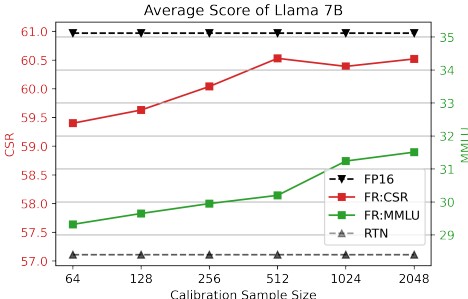

Figure 2: Zero-shot and five-shot accuracies of Llama 7B for FlexRound (FR) on common sense reasoning (CSR) tasks and MMLU according to the calibration sample size, with 8-bit per-channel asymmetric weight quantization and 8-bit *per-tensor* asymmetric *static* activation quantization.

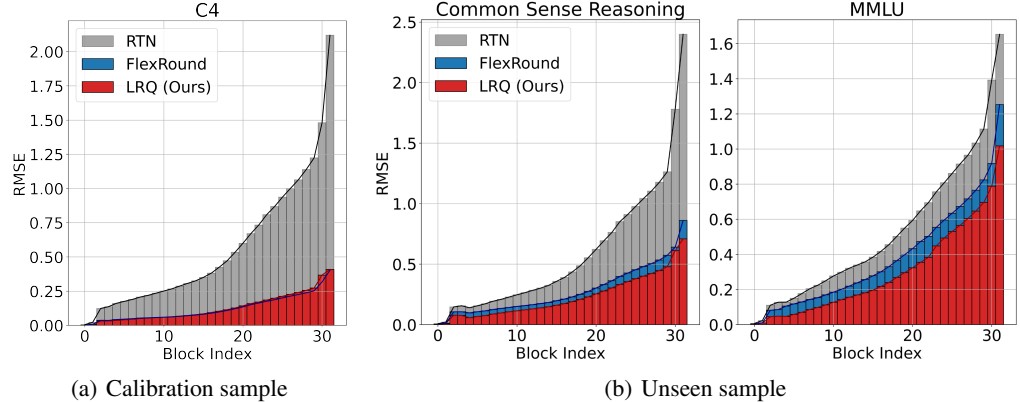

(a) Calibration sample

(b) Unseen sample

Figure 3: Accumulated root mean square error (RMSE) between $\boldsymbol{W}\boldsymbol{X}$ and $\widehat{\boldsymbol{W}}\widetilde{\boldsymbol{X}}$ for RTN, FlexRound, and LRQ on (a) a calibration sample from the C4 dataset and (b) an unseen sample from common sense reasoning and MMLU benchmarks, ranging from the first Transformer block to the last Transformer block of Llama 7B. Here, weights and activations are quantized to 8-bit with per-channel asymmetric quantization and *per-tensor* asymmetric *static* quantization, respectively. Note that RMSE tends to rise in line with the block index due to the presence of $\widetilde{\boldsymbol{X}}$ that accumulates quantization error resulting from previous quantized Transformer blocks.

as the calibration sample size grows larger. Unfortunately, however, FlexRound still falls behind the FP16 baseline on MMLU by more than 3.5 percent even utilizing 2048 calibration samples, the maximum number we can use on a single NVIDIA A100-80GB GPU during the block-wise reconstruction process. Hence, we turn our focus toward reducing the number of learnable parameters.

## 2.3 LOW-RANK QUANTIZATION

To reduce the number of learnable parameters, we decompose a weight-scaling matrix, $\boldsymbol{S}_2$, into a low-rank matrix before performing the reconstruction process. To be more specific, for $\boldsymbol{W} \in \mathbb{R}^{C_{out} \times C_{in}}$, $\boldsymbol{S}_2 \in \mathbb{R}^{C_{out} \times C_{in}}$ is factorized into $\boldsymbol{L}_2 \boldsymbol{U}_2$ where $\boldsymbol{L}_2 \in \mathbb{R}^{C_{out} \times r}$ and $\boldsymbol{U}_2 \in \mathbb{R}^{r \times C_{in}}$ for $r < \min(C_{out}, C_{in})$. Additionally, we supplement $\boldsymbol{L}_2 \boldsymbol{U}_2$ with a row vector, $\boldsymbol{r}_2 \in \mathbb{R}^{C_{out} \times 1}$, and a column vector, $\boldsymbol{c}_2 \in \mathbb{R}^{1 \times C_{in}}$, which is inspired by the addition of a row or column vector (or both) to a low-rank matrix in recommendation systems, one of the most popular applications of low-rank structure, for better prediction of ratings by considering a bias for each user or each item (Jahrer & Töscher, 2012; Goodfellow et al., 2016; Koren et al., 2021). As a result, we formulate $\widehat{\boldsymbol{W}}$ as

$$\widehat{\boldsymbol{W}} = \boldsymbol{s}_1 \left\lfloor \frac{\boldsymbol{W}}{\boldsymbol{s}_1 \odot \exp(\boldsymbol{L}_2 \boldsymbol{U}_2 + \boldsymbol{r}_2 + \boldsymbol{c}_2)} \right\rceil, \tag{2}$$

which we refer to as 'Low-Rank Quantization (LRQ)'. At first, $\boldsymbol{L}_2$ and $\boldsymbol{U}_2$ are initialized to zeros and random values from a normal distribution respectively, and $\boldsymbol{r}_2$ and $\boldsymbol{c}_2$ are set to zero vectors so that $\boldsymbol{L}_2 \boldsymbol{U}_2 + \boldsymbol{r}_2 + \boldsymbol{c}_2$ starts from a zero matrix like $\boldsymbol{S}_2$ in Eq. 1. Then, $\boldsymbol{s}_1$, $\boldsymbol{L}_2$, $\boldsymbol{U}_2$, $\boldsymbol{r}_2$, and $\boldsymbol{c}_2$ are learned to minimize $\|\boldsymbol{W}\boldsymbol{X} - \widehat{\boldsymbol{W}}\widetilde{\boldsymbol{X}}\|_2^2$ in a block-by-block manner.

## 2.4 EFFECT OF LOW-RANK MATRICES ON GENERALIZATION ABILITY OF QUANTIZED LLMS

Considering that a full weight-scaling matrix is substituted with a low-rank matrix as seen in Eq. 2 derived from Eq. 1, one might wonder (i) whether the minimization of block reconstruction error on calibration samples is feasible despite the use of low-rank matrices, and (ii) how the utilization of low-rank matrices can result in improved generalization performance on unseen benchmarks as Figure 1 demonstrates. To address these concerns, we conduct a comparative analysis of accumulated root mean square error (RMSE) between $\boldsymbol{W}\boldsymbol{X}$ and $\widehat{\boldsymbol{W}}\widetilde{\boldsymbol{X}}$ for RTN, FlexRound, and LRQ.

For a calibration sample that is selected from the C4 dataset, even if both FlexRound and LRQ initially start their learning process from the same RTN baseline, LRQ achieves an almost identical accumulated RMSE to FlexRound, as illustrated in Figure 3(a). This observation underscores that the use of low-rank weight-scaling matrices does not pose any noticeable obstacle to the minimization of

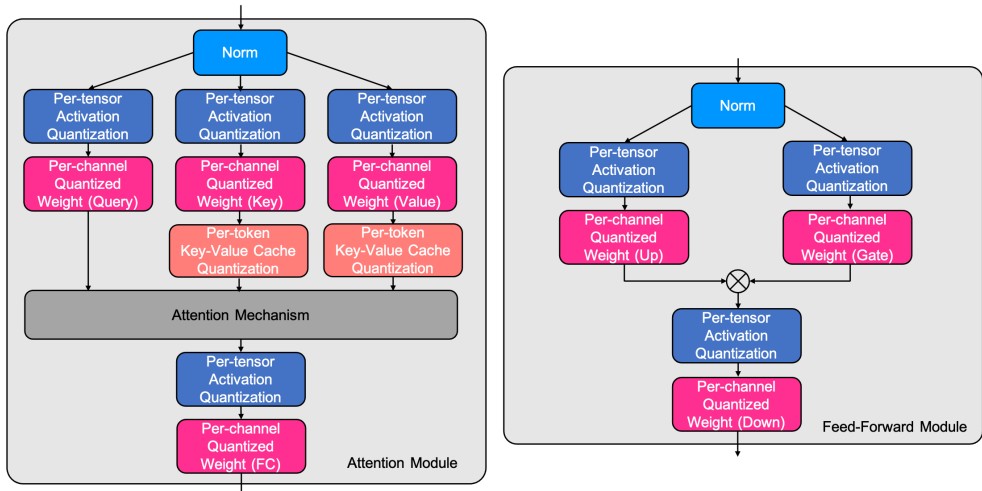

Figure 4: Illustration of a quantized Transformer block with per-channel asymmetric weight quanti-zation, *per-tensor* asymmetric *static* activation quantization, and per-token asymmetric KV cache quantization. We remain the inputs of softmax and normalization layers in FP16.

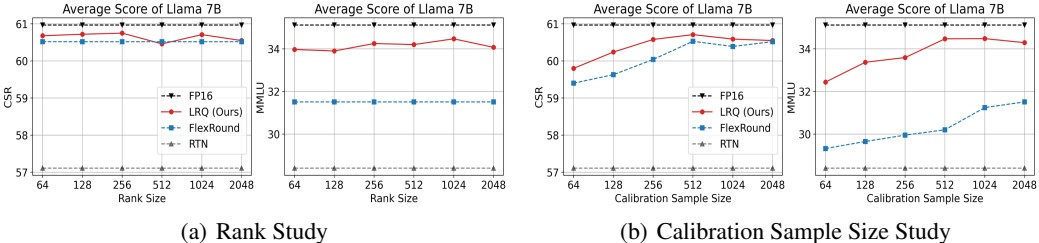

| (a) Rank Study | (b) Calibration Sample Size Study |

Figure 5: Zero-shot and five-shot performances of Llama 7B on common sense reasoning (CSR) tasks and MMLU respectively, where weights and activations are quantized to 8-bit as described in Figure 4 while the KV cache is kept in FP16.

block reconstruction error on calibration data. For common sense reasoning and MMLU benchmarks that are unseen during the reconstruction stage, however, accumulated RMSE for LRQ is much smaller than that for FlexRound as well as RTN as described in Figure 3(b). This compelling result implies that harnessing the parameter-efficiency of low-rank matrices can facilitate superior generalization on unseen benchmarks. In light of these findings, the incorporation of low-rank matrices into block-wise reconstruction is indeed a pivotal step in enhancing the generalization capability of quantized LLMs.

## 3 EXPERIMENTS

In this section, we first explore the influence of the rank $r$ in Eq. 2 and the quantity of calibration samples on the performance of LRQ. Next, to verify the effectiveness of LRQ, we compare LRQ with existing state-of-the-art post-training quantization (PTQ) methods for open-source large language models (LLMs) such as Llama and Llama 2 by adopting per-channel asymmetric weight quantization, per-tensor asymmetric static activation quantization, and per-token asymmetric KV cache quantization, as illustrated in Figure 4. For the Llama 2 models in Table 2, however, the accuracy gap on the massive multitask language understanding (MMLU) benchmark between quantized LLMs and their FP16 baselines is observed. In this sense, we also perform experiments on Llama 2 by switching from per-tensor asymmetric static activation quantization to per-token asymmetric activation quantization.

We use just a single NVIDIA A100-80GB GPU to quantize all the Llama and Llama 2 models via LRQ. We randomly choose 512 calibration samples with a token length of 1024 from the training set of C4 (Raffel et al., 2020), one of the pre-training datasets for Llama. Unless otherwise mentioned, LRQ is applied to all linear layers in both attention and feed-forward modules, and the rank $r$ in Eq. 2 is set to 2048 for large language models beyond 30B parameters or to 1024 for smaller models in order to reduce the number of learnable parameters by approximately half compared to FlexRound.

Table 1: Zero-shot performance of Llama and Llama 2 on common sense reasoning tasks (BoolQ, PIQA, HellaSwag, WinoGrande, ARC easy and challenge, and OpenBookQA) with per-channel asymmetric weight quantization, per-tensor asymmetric static activation quantization, and per-token asymmetric KV cache quantization. The accuracy (%) is reported for all tasks. The number of bits used for weights, activations, and KV cache is expressed as W/A/KV.

| Method | # Bits (W/A/KV) | BoolQ | PIQA | HellaSwag | WinoGrande | ARC-e | ARC-c | OBQA | Average |
|---|---|---|---|---|---|---|---|---|---|
| Llama 7B | 16/16/16 | 73.15 | 77.31 | 72.96 | 67.09 | 52.48 | 41.38 | 42.40 | 60.97 |
| RTN | 8/8/8 | 69.76 | 73.72 | 65.95 | 62.75 | 48.91 | 37.12 | 37.60 | 56.54 |
| SmoothQuant | 8/8/8 | 69.42 | 72.63 | 69.07 | 64.72 | 48.61 | 37.12 | 39.20 | 57.25 |
| FlexRound | 8/8/8 | 72.54 | 76.50 | 71.88 | 66.77 | 53.03 | 39.76 | 42.00 | 60.35 |
| LRQ (Ours) | 8/8/8 | 72.84 | 77.37 | 72.04 | 67.01 | 53.03 | 40.53 | 41.60 | **60.63** |
| Llama 13B | 16/16/16 | 68.53 | 79.11 | 76.23 | 70.01 | 59.89 | 44.54 | 42.20 | 62.93 |
| RTN | 8/8/8 | 65.87 | 72.25 | 62.52 | 62.19 | 51.81 | 35.41 | 38.40 | 55.49 |
| SmoothQuant | 8/8/8 | 67.34 | 75.19 | 71.78 | 69.06 | 54.92 | 40.44 | 38.80 | 59.65 |
| FlexRound | 8/8/8 | 68.78 | 78.51 | 75.23 | 70.56 | 58.46 | 44.03 | 41.00 | 62.37 |
| LRQ (Ours) | 8/8/8 | 68.84 | 78.78 | 75.56 | 70.80 | 59.13 | 44.62 | 41.60 | **62.76** |
| Llama 33B | 16/16/16 | 68.38 | 80.09 | 79.21 | 72.93 | 58.92 | 45.48 | 42.00 | 63.86 |
| RTN | 8/8/8 | 68.81 | 76.55 | 68.76 | 66.06 | 56.48 | 42.49 | 42.40 | 60.22 |
| SmoothQuant | 8/8/8 | 71.31 | 75.30 | 71.29 | 68.98 | 53.66 | 43.26 | 41.00 | 60.69 |
| FlexRound | 8/8/8 | 69.05 | 79.49 | 77.49 | 70.88 | 56.86 | 43.60 | 42.00 | 62.77 |
| LRQ (Ours) | 8/8/8 | 68.84 | 79.98 | 78.52 | 73.72 | 58.21 | 45.73 | 43.00 | **64.00** |
| Llama 65B | 16/16/16 | 82.32 | 80.85 | 80.71 | 77.19 | 58.71 | 46.33 | 44.60 | 67.24 |
| RTN | 8/8/8 | 79.51 | 75.79 | 74.13 | 71.35 | 51.85 | 44.03 | 43.60 | 62.89 |
| SmoothQuant | 8/8/8 | 78.78 | 79.54 | 79.11 | 73.32 | 56.23 | 45.90 | 43.80 | 65.24 |
| FlexRound | 8/8/8 | 80.46 | 79.38 | 79.23 | 74.98 | 57.20 | 46.42 | 45.00 | 66.10 |
| LRQ (Ours) | 8/8/8 | 82.35 | 81.12 | 79.96 | 75.61 | 58.96 | 46.59 | 45.40 | **67.14** |
| Llama 2 7B | 16/16/16 | 71.07 | 76.99 | 72.96 | 67.25 | 53.58 | 40.53 | 40.80 | 60.45 |
| RTN | 8/8/8 | 60.58 | 67.08 | 57.66 | 60.54 | 45.83 | 31.57 | 34.40 | 51.09 |
| SmoothQuant | 8/8/8 | 67.09 | 73.29 | 67.26 | 64.25 | 51.18 | 36.26 | 38.40 | 56.82 |
| FlexRound | 8/8/8 | 72.05 | 77.26 | 71.30 | 65.98 | 54.88 | 39.16 | 39.20 | **59.98** |
| LRQ (Ours) | 8/8/8 | 67.86 | 76.99 | 71.97 | 67.01 | 54.71 | 40.19 | 40.00 | 59.82 |
| Llama 2 13B | 16/16/16 | 69.02 | 79.05 | 76.62 | 69.61 | 57.95 | 44.28 | 42.00 | 62.65 |
| RTN | 8/8/8 | 62.97 | 73.72 | 62.60 | 57.77 | 52.86 | 36.77 | 37.00 | 54.81 |
| SmoothQuant | 8/8/8 | 63.94 | 74.97 | 70.50 | 65.43 | 54.88 | 40.78 | 38.60 | 58.44 |
| FlexRound | 8/8/8 | 66.94 | 79.00 | 75.32 | 69.38 | 58.54 | 42.92 | 40.40 | 61.79 |
| LRQ (Ours) | 8/8/8 | 68.59 | 78.67 | 75.83 | 70.64 | 58.16 | 43.34 | 39.80 | **62.15** |
| Llama 2 70B | 16/16/16 | 76.70 | 80.85 | 80.85 | 76.95 | 59.72 | 47.95 | 44.40 | 66.77 |
| RTN | 8/8/8 | 72.39 | 78.51 | 76.49 | 69.61 | 57.74 | 44.62 | 40.40 | 62.82 |
| SmoothQuant | 8/8/8 | 76.73 | 77.15 | 79.37 | 73.01 | 55.56 | 46.59 | 43.20 | 64.52 |
| FlexRound | 8/8/8 | 76.18 | 80.36 | 79.09 | 75.06 | 60.10 | 46.42 | 43.80 | 65.86 |
| LRQ (Ours) | 8/8/8 | 77.95 | 81.23 | 79.78 | 74.82 | 57.83 | 46.33 | 43.60 | **65.93** |

All quantized models are evaluated on MMLU (Hendrycks et al., 2021) in the five-shot setting as well as six common sense reasoning benchmarks: BoolQ (Clark et al., 2019), PIQA (Bisk et al., 2020), HellaSwag (Zellers et al., 2019), WinoGrande (Sakaguchi et al., 2021), ARC easy and challenge (Clark et al., 2018), and OpenBookQA (Mihaylov et al., 2018) in the zero-shot setting.

## 3.1 ABLATION STUDY

**Rank Study** To examine the impact of the rank $r$ in Eq. 2 on the generalization on unseen benchmarks, we compare LRQ with different $r$ to FlexRound for Llama 7B as shown in Figure 5(a). For common sense reasoning tasks, both FlexRound and LRQ show negligible accuracy degradation compared to the FP16 baseline. When it comes to MMLU, however, LRQ significantly outperforms FlexRound regardless of $r$ despite the fact that LRQ utilizes 512 calibration samples whereas FlexRound exploits 2048 samples. Such a comparison is particularly noteworthy because FlexRound has a tendency to exhibit improved performance with an increase in the calibration sample size as discussed in Section 2.2.

**Calibration Sample Size Study** To identify whether the performance of LRQ does improve with an increase in the number of calibration samples, we also conduct experiments on LRQ for Llama

Table 2: Five-shot accuracy of Llama and Llama 2 on Massive Multitask Language Understanding with per-channel asymmetric weight quantization, per-tensor asymmetric static activation quantization, and per-token asymmetric KV cache quantization. The accuracy (%) is reported for four disciplines. The number of bits used for weights, activations, and KV cache is expressed as W/A/KV.

| Method | # Bits (W/A/KV) | STEM | Humanities | Social Science | Other | Average |
|---|---|---|---|---|---|---|
| Llama 7B | 16/16/16 | 30.58 | 33.88 | 38.19 | 38.25 | 35.12 |
| RTN | 8/8/8 | 27.04 | 27.23 | 29.28 | 30.38 | 28.36 |
| SmoothQuant | 8/8/8 | 28.40 | 28.69 | 32.79 | 30.48 | 29.94 |
| FlexRound | 8/8/8 | 27.60 | 28.71 | 29.61 | 31.99 | 29.43 |
| LRQ (Ours) | 8/8/8 | 29.72 | 32.79 | 37.44 | 38.16 | **34.39** |
| Llama 13B | 16/16/16 | 36.35 | 44.97 | 54.14 | 53.15 | 47.02 |
| RTN | 8/8/8 | 26.38 | 25.33 | 27.95 | 24.83 | 26.01 |
| SmoothQuant | 8/8/8 | 27.24 | 30.12 | 30.58 | 31.31 | 29.87 |
| FlexRound | 8/8/8 | 33.63 | 42.81 | 48.65 | 49.26 | 43.60 |
| LRQ (Ours) | 8/8/8 | 35.16 | 44.55 | 51.74 | 52.04 | **45.83** |
| Llama 33B | 16/16/16 | 46.69 | 56.39 | 67.40 | 63.60 | 58.38 |
| RTN | 8/8/8 | 32.47 | 32.37 | 38.35 | 40.59 | 35.60 |
| SmoothQuant | 8/8/8 | 37.94 | 41.64 | 50.57 | 51.48 | 45.07 |
| FlexRound | 8/8/8 | 43.47 | 52.20 | 61.94 | 59.90 | 54.24 |
| LRQ (Ours) | 8/8/8 | 45.26 | 52.58 | 63.99 | 61.26 | **55.51** |
| Llama 65B | 16/16/16 | 51.95 | 61.87 | 73.32 | 67.58 | 63.57 |
| RTN | 8/8/8 | 41.22 | 47.23 | 61.39 | 54.69 | 50.76 |
| SmoothQuant | 8/8/8 | 44.83 | 50.82 | 63.34 | 57.09 | 53.72 |
| FlexRound | 8/8/8 | 46.32 | 54.60 | 65.06 | 62.49 | 56.94 |
| LRQ (Ours) | 8/8/8 | 50.96 | 61.28 | 71.99 | 66.66 | **62.65** |
| Llama 2 7B | 16/16/16 | 37.04 | 43.38 | 51.84 | 52.44 | 45.96 |
| RTN | 8/8/8 | 29.66 | 24.06 | 30.45 | 24.49 | 26.76 |
| SmoothQuant | 8/8/8 | 21.67 | 25.06 | 22.26 | 24.03 | 23.48 |
| FlexRound | 8/8/8 | 33.40 | 36.96 | 43.13 | 46.30 | 39.70 |
| LRQ (Ours) | 8/8/8 | 34.82 | 39.91 | 46.47 | 47.62 | **42.04** |
| Llama 2 13B | 16/16/16 | 44.27 | 54.43 | 63.41 | 60.76 | 55.68 |
| RTN | 8/8/8 | 29.06 | 24.23 | 29.93 | 29.03 | 27.62 |
| SmoothQuant | 8/8/8 | 21.31 | 24.08 | 21.71 | 23.72 | 22.88 |
| FlexRound | 8/8/8 | 41.09 | 51.58 | 61.39 | 59.41 | 53.28 |
| LRQ (Ours) | 8/8/8 | 42.88 | 51.97 | 62.14 | 59.93 | **54.08** |
| Llama 2 70B | 16/16/16 | 57.79 | 65.16 | 80.44 | 74.61 | 69.11 |
| RTN | 8/8/8 | 46.82 | 53.37 | 66.23 | 58.51 | 55.97 |
| SmoothQuant | 8/8/8 | 47.28 | 54.60 | 69.32 | 63.33 | 58.27 |
| FlexRound | 8/8/8 | 54.27 | 61.11 | 77.45 | 71.31 | 65.57 |
| LRQ (Ours) | 8/8/8 | 54.44 | 62.61 | 76.99 | 71.78 | **66.12** |

7B with various calibration sample size while fixing $r$ to 1024. The accuracy of LRQ does rise with a larger calibration sample size, but it reaches a saturation point when exceeding 1024 calibration samples as depicted in Figure 5(b). Nevertheless, LRQ can surpass FlexRound irrespective of the calibration sample size not only on common sense reasoning benchmarks but also on the MMLU benchmark, which sheds light on the effect of low-rank matrices on enhancing generalization in quantized LLMs as we elaborate on in Section 2.4.

## 3.2 PER-TENSOR ASYMMETRIC STATIC ACTIVATION QUANTIZATION

As meticulously studied in Xiao et al. (2022), per-tensor static activation quantization is hardware-efficient and can be implemented on off-the-shelf GPUs with FasterTransformer, the state-of-the-art Transformer inference framework provided from NVIDIA, to achieve up to $1.5\times$ inference speed-up and almost halving the memory footprint compared to FP16 baselines. Accordingly, we employ

Table 3: Zero-shot performance of Llama 2 on common sense reasoning tasks (BoolQ, PIQA, HellaSwag, WinoGrande, ARC easy and challenge, and OpenBookQA) with per-channel asymmetric weight quantization, per-token asymmetric activation quantization, and per-token asymmetric KV cache quantization. The accuracy (%) is reported for all tasks. The number of bits used for weights, activations, and KV cache is expressed as W/A/KV.

| Method | # Bits (W/A/KV) | BoolQ | PIQA | HellaSwag | WinoGrande | ARC-e | ARC-c | OBQA | Average |
|---|---|---|---|---|---|---|---|---|---|
| Llama 2 7B | 16/16/16 | 71.07 | 76.99 | 72.96 | 67.25 | 53.58 | 40.53 | 40.80 | 60.45 |
| RTN | 8/8/8 | 69.60 | 77.20 | 72.26 | 67.09 | 53.62 | 39.85 | 41.00 | 60.09 |
| SmoothQuant | 8/8/8 | 70.73 | 76.77 | 72.80 | 67.25 | 53.70 | 41.04 | 40.60 | 60.41 |
| FlexRound | 8/8/8 | 72.02 | 77.09 | 72.50 | 67.40 | 54.17 | 40.19 | 40.80 | 60.60 |
| LRQ (Ours) | 8/8/8 | 72.45 | 77.04 | 72.70 | 67.09 | 53.66 | 40.61 | 41.60 | **60.74** |
| RTN | 4/8/8 | 68.13 | 75.14 | 65.89 | 62.67 | 46.42 | 36.52 | 36.20 | 55.85 |
| SmoothQuant | 4/8/8 | 53.27 | 69.64 | 51.28 | 56.35 | 40.95 | 31.83 | 35.00 | 48.33 |
| FlexRound | 4/8/8 | 71.71 | 76.77 | 72.24 | 66.14 | 53.49 | 40.02 | 40.40 | 60.11 |
| LRQ (Ours) | 4/8/8 | 73.00 | 76.99 | 71.90 | 65.98 | 54.38 | 39.68 | 41.20 | **60.45** |
| Llama 2 13B | 16/16/16 | 69.02 | 79.05 | 76.62 | 69.61 | 57.95 | 44.28 | 42.00 | 62.65 |
| RTN | 8/8/8 | 67.46 | 78.73 | 75.57 | 68.51 | 58.12 | 44.28 | 41.40 | 62.01 |
| SmoothQuant | 8/8/8 | 69.33 | 79.05 | 76.52 | 69.38 | 58.04 | 44.37 | 42.00 | 62.67 |
| FlexRound | 8/8/8 | 69.36 | 79.16 | 76.67 | 69.53 | 57.83 | 44.37 | 42.80 | **62.82** |
| LRQ (Ours) | 8/8/8 | 69.02 | 78.78 | 76.48 | 69.93 | 57.83 | 43.86 | 42.00 | 62.56 |
| RTN | 4/8/8 | 65.23 | 74.05 | 60.04 | 58.64 | 49.07 | 35.92 | 35.20 | 54.02 |
| SmoothQuant | 4/8/8 | 62.17 | 61.37 | 44.59 | 53.67 | 36.99 | 28.24 | 32.40 | 45.63 |
| FlexRound | 4/8/8 | 69.05 | 78.51 | 75.51 | 69.53 | 58.75 | 43.60 | 41.20 | 62.31 |
| LRQ (Ours) | 4/8/8 | 71.13 | 78.29 | 75.79 | 68.90 | 57.83 | 43.34 | 41.20 | **62.35** |
| Llama 2 70B | 16/16/16 | 76.70 | 80.85 | 80.85 | 76.95 | 59.72 | 47.95 | 44.40 | 66.77 |
| RTN | 8/8/8 | 76.02 | 81.07 | 80.45 | 75.61 | 60.31 | 47.87 | 43.80 | 66.45 |
| SmoothQuant | 8/8/8 | 76.15 | 80.96 | 80.63 | 77.11 | 59.09 | 47.87 | 44.60 | 66.63 |
| FlexRound | 8/8/8 | 75.93 | 81.45 | 80.48 | 75.85 | 60.06 | 48.55 | 44.80 | 66.73 |
| LRQ (Ours) | 8/8/8 | 75.99 | 81.50 | 80.61 | 75.77 | 59.97 | 49.49 | 45.20 | **66.93** |
| RTN | 4/8/8 | 75.90 | 79.22 | 71.39 | 70.56 | 53.11 | 43.60 | 40.40 | 62.03 |
| SmoothQuant | 4/8/8 | 66.24 | 75.84 | 56.19 | 60.46 | 50.25 | 36.01 | 40.40 | 55.06 |
| FlexRound | 4/8/8 | 77.31 | 80.96 | 79.89 | 75.30 | 60.19 | 48.21 | 43.40 | 66.47 |
| LRQ (Ours) | 4/8/8 | 77.92 | 81.28 | 80.42 | 75.06 | 60.94 | 48.04 | 42.60 | **66.61** |

per-tensor asymmetric static activation quantization as well as per-channel asymmetric weight quantization. Moreover, we also quantize the KV cache to 8-bit with a per-token asymmetric quantization scheme. It is worth noting that for large batch sizes, the KV cache can consume a much larger amount of memory than the model size, thus causing a bottleneck in high-throughput LLM inference. Fortunately, the performance discrepancy before and after per-token asymmetric KV cache quantization is almost insignificant no matter which quantization method is selected, as presented in Appendix B. For these reasons, we utilize per-channel asymmetric weight quantization, per-tensor asymmetric static activation quantization, and per-token asymmetric KV cache quantization as exemplified in Figure 4. Further experimental details are provided in Appendix C.

Table 1 and 2 reveal the effectiveness of LRQ compared with the state-of-the-art LLM PTQ techniques on common sense reasoning tasks and MMLU, respectively. For common sense reasoning bechmarks, the zero-shot performance of LRQ is almost close to that of FP16 baselines, being superior to that of both SmoothQuant and FlexRound for most of the Llama and Llama 2 models. Not only that, LRQ also considerably outperforms SmoothQuant and FlexRound on MMLU.

## 3.3 PER-TOKEN ASYMMETRIC ACTIVATION QUANTIZATION

Although LRQ shows better performance than SmoothQuant and FlexRound on both common sense reasoning tasks and MMLU when employing per-channel asymmetric weight quantization, per-tensor asymmetric static activation quantization and per-token asymmetric KV cache quantization, there is still the five-shot performance gap on MMLU between LRQ and FP16 baselines for Llama 2 as shown in Table 2. Because of this, we also conduct experiments on Llama 2 with a per-token asymmetric activation quantization scheme instead of a per-tensor asymmetric static activation quantization scheme. More details about experimental settings are given in Appendix C.

In Table 3 and 4, activations and KV cache are quantized to 8-bit with a per-token asymmetric quantization scheme. In the case of 8-bit per-channel asymmetric weight quantization, although

Table 4: Five-shot accuracy of Llama 2 on Massive Multitask Language Understanding with per-channel asymmetric weight quantization, per-token asymmetric activation quantization, and per-token asymmetric KV cache quantization. The accuracy (%) is reported for four disciplines. The number of bits used for weights, activations, and KV cache is expressed as W/A/KV.

| Method | # Bits (W/A/KV) | STEM | Humanities | Social Science | Other | Average |
|---|---|---|---|---|---|---|
| Llama 2 7B | 16/16/16 | 37.04 | 43.38 | 51.84 | 52.44 | 45.96 |
| RTN | 8/8/8 | 36.15 | 42.85 | 50.34 | 52.31 | 45.24 |
| SmoothQuant | 8/8/8 | 36.98 | 42.93 | 51.87 | 52.56 | 45.83 |
| FlexRound | 8/8/8 | 36.98 | 42.91 | 51.87 | 52.28 | 45.76 |
| LRQ (Ours) | 8/8/8 | 36.88 | 43.12 | 51.67 | 52.53 | **45.83** |
| RTN | 4/8/8 | 28.00 | 25.80 | 27.53 | 28.01 | 27.16 |
| SmoothQuant | 4/8/8 | 25.75 | 24.91 | 22.49 | 26.59 | 24.95 |
| FlexRound | 4/8/8 | 37.81 | 42.55 | 50.47 | 50.65 | 45.14 |
| LRQ (Ours) | 4/8/8 | 36.88 | 42.53 | 50.80 | 52.22 | **45.36** |
| Llama 2 13B | 16/16/16 | 44.27 | 54.43 | 63.41 | 60.76 | 55.68 |
| RTN | 8/8/8 | 43.87 | 52.88 | 62.33 | 60.67 | 54.81 |
| SmoothQuant | 8/8/8 | 43.74 | 53.20 | 63.18 | 60.83 | 55.11 |
| FlexRound | 8/8/8 | 44.17 | 52.88 | 63.76 | 61.29 | 55.33 |
| LRQ (Ours) | 8/8/8 | 44.50 | 53.07 | 63.24 | 61.26 | **55.35** |
| RTN | 4/8/8 | 30.95 | 26.31 | 32.92 | 34.58 | 30.67 |
| SmoothQuant | 4/8/8 | 27.87 | 24.95 | 26.58 | 27.91 | 26.62 |
| FlexRound | 4/8/8 | 42.88 | 50.71 | 61.94 | 59.93 | 53.77 |
| LRQ (Ours) | 4/8/8 | 43.90 | 52.56 | 62.07 | 59.96 | **54.49** |
| Llama 2 70B | 16/16/16 | 57.79 | 65.16 | 80.44 | 74.61 | 69.11 |
| RTN | 8/8/8 | 56.23 | 63.55 | 78.39 | 73.01 | 67.41 |
| SmoothQuant | 8/8/8 | 57.59 | 64.21 | 80.70 | 74.58 | **68.79** |
| FlexRound | 8/8/8 | 57.22 | 63.97 | 79.62 | 73.81 | 68.22 |
| LRQ (Ours) | 8/8/8 | 57.95 | 63.85 | 80.34 | 73.94 | 68.52 |
| RTN | 4/8/8 | 41.19 | 45.74 | 57.52 | 53.61 | 49.16 |
| SmoothQuant | 4/8/8 | 29.69 | 31.31 | 36.89 | 37.42 | 33.59 |
| FlexRound | 4/8/8 | 56.26 | 62.89 | 78.78 | 72.92 | 67.26 |
| LRQ (Ours) | 4/8/8 | 55.57 | 64.65 | 78.97 | 72.52 | **67.65** |

the accuracy of LRQ is slightly higher than that of SmoothQuant and FlexRound for most Llama 2 models, it can be concluded that SmoothQuant, FlexRound, and LRQ are almost evenly matched. More interestingly, rounding-to-nearest (RTN) also performs nearly to the levels of FP16 baselines. As a consequence, we reduce the weight bits to 4-bit for a further reduction in the model size. Surprisingly, even when quantizing weights to 4-bit and both activations and KV cache to 8-bit, LRQ can attain similar zero-shot performance to FP16 baselines on common sense reasoning benchmarks and narrow the five-shot performance difference between FP16 baselines and quantized models to less than 1.5 percent on the MMLU benchmark. We for the first time carry out experiments on 4-bit weight, 8-bit activation, and 8-bit KV cache quantization, demonstrating that LRQ exhibits the minimal performance degradation among LLM PTQ methods.

## 4 CONCLUSION

We propose a simple yet effective post-training weight quantization method for LLMs, *LRQ* that learns low-rank weight-scaling matrices for block-by-block reconstructing the outputs of an intermediate Transformer block. Thanks to the use of such low-rank matrices, we can decrease the number of learnable parameters effectively while allowing for scaling weights individually due to the sharing of learnable parameters through a low-rank structure, thereby enhancing the generalization performance of quantized LLMs. Through comprehensive experiments, we demonstrate the superiority of LRQ over existing LLM post-training weight-activation quantization approaches. Notably, we are the first to run experiments on 4-bit weight and 8-bit activation quantization with minimal accuracy drop among LLM post-training quantization (PTQ) techniques. We hope that this noticeable result would pave the way for the possibility of 4-bit weight and 4-bit activation quantization for LLMs via PTQ.

## REPRODUCIBILITY STATEMENT

To make it possible to reproduce our experimental results in this paper, we delineate experimental setups in Appendix C in as much detail as possible. However, we are planning to release our code as soon as possible in the near future in order to encourage many LLM engineers and researchers to benefit from our method, LRQ.

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

# A    RELATED WORK

Quantization works can be generally categorized into quantization-aware training (QAT) and post-training quantization (PTQ). As QAT can maintain the performance of FP32/FP16 baselines, QAT has been applied to computer vision models (Jung et al., 2019; Esser et al., 2020; Lee et al., 2021; Kim et al., 2021). Notwithstanding, there exist many challenges associated with applying QAT to large language models (LLMs) due to the sheer scale of pre-training data and a huge amount of computing resources required for training on the whole pre-training dataset. Although Liu et al. (2023a) presented the possibility of applying QAT to LLMs, unforunately, they did not perform experiments on Llama 65B, the largest and best performing model among the Llama models, in spite of using a single 8-GPU node. On the other hand, as Frantar et al. (2023) demonstrated the application of PTQ to LLMs only with a single GPU, many researchers have recently paid attention to PTQ for LLMs.

LLM PTQ can be classified into two categories: LLM weight-only quantization (Frantar et al., 2023; Lin et al., 2023) and LLM weight-activation quantization (Dettmers et al., 2022; Yao et al., 2022; Xiao et al., 2022; Lee et al., 2023). For the former quantization, Frantar et al. (2023) quantized the weights of LLMs into low-bit integers based on layer-wise reconstruction, whereas Lin et al. (2023) did by not counting on reconstruction but per-channel scaling in consideration of both weight and activation magnitudes. Despite the fact that both studies exhibited decent quantization performance, the main benefit of weight-only quantization does not align with serving LLMs with high throughput as delineated in Section 1. In this light, we concentrates on weight-activation quantization.

When it comes to weight-activation quantization, Yao et al. (2022) presented ZeroQuant with a group-wise weight quantization scheme and a per-token activation quantization scheme based on layer-wise knowledge distillation, and Dettmers et al. (2022) proposed LLM.int8() with a per-channel weight quantization scheme and a per-token activation quantization scheme while keeping activation outliers in FP16. As discussed in Xiao et al. (2022), however, ZeroQuant incurs severe accuracy degradation for an open-source LLM, and the inference latency of LLM.int8() can be higher than that of the FP16 baseline. To deal with both issues, Xiao et al. (2022) devised SmoothQuant that can preserve the accuracy of OPT (Zhang et al., 2022) by easing the difficulty of activation quantization and accelerate LLM inference by up to $1.5$ times. Yet, SmoothQuant suffers from non-negligible performance degradation for other open-source models such as Llama and Llama 2 with a per-tensor static activation quantization scheme as illustrated in Figure 1. FlexRound that Lee et al. (2023) created showed the experimental results of Llama up to 33B with a per-channel weight quantization scheme and a per-tensor static activation quantization scheme, but FlexRound incurs considerable performance degradation on the massive multitask language understanding (MMLU) benchmark as described in Figure 1(b).

# B  COMPARISON OF EXPERIMENTAL RESULTS BEFORE AND AFTER PER-TOKEN ASYMMETRIC KV CACHE QUANTIZATION

Table 5, 6, 7, 8, 9, and 10 show the comparison of experimental results before and after per-token asymmetric KV cache quantization. It can be easily seen that the performance difference before and after per-token asymmetric KV cache quantization is nearly inconsiderable no matter which quantization technique is chosen, as mentioned in Section 3.2. Furthermore, even without per-token asymmetric KV cache quantization, LRQ still outperforms prior state-of-the-art LLM post-training weight-activation quantization methods in most cases.

Table 5: Zero-shot performance of Llama on common sense reasoning tasks (BoolQ, PIQA, HellaSwag, WinoGrande, ARC easy and challenge, and OpenBookQA) with per-channel asymmetric weight quantization, per-tensor asymmetric static activation quantization, and per-token asymmetric KV cache quantization (if applied). Please refer to Figure 4. The accuracy (%) is reported for common sense reasoning tasks. The number of bits used for weights, activations, and KV cache is expressed as W/A/KV.

| Method | # Bits (W/A/KV) | BoolQ | PIQA | HellaSwag | WinoGrande | ARC-e | ARC-c | OBQA | Average |
|--------|-----------------|-------|------|-----------|------------|-------|-------|------|---------|
| Llama 7B | 16/16/16 | 73.15 | 77.31 | 72.96 | 67.09 | 52.48 | 41.38 | 42.40 | 60.97 |
| RTN | 8/8/16 | 71.56 | 73.72 | 65.86 | 63.93 | 49.49 | 36.43 | 38.80 | 57.11 |
| SmoothQuant | 8/8/16 | 69.63 | 73.12 | 68.88 | 65.43 | 48.70 | 38.57 | 38.00 | 57.48 |
| FlexRound | 8/8/16 | 73.76 | 76.66 | 71.75 | 67.01 | 52.31 | 40.02 | 42.20 | 60.53 |
| LRQ (Ours) | 8/8/16 | 73.03 | 77.64 | 72.10 | 66.77 | 52.95 | 40.87 | 41.60 | **60.71** |
| RTN | 8/8/8 | 69.76 | 73.72 | 65.95 | 62.75 | 48.91 | 37.12 | 37.60 | 56.54 |
| SmoothQuant | 8/8/8 | 69.42 | 72.63 | 69.07 | 64.72 | 48.61 | 37.12 | 39.20 | 57.25 |
| FlexRound | 8/8/8 | 72.54 | 76.50 | 71.88 | 66.77 | 53.03 | 39.76 | 42.00 | 60.35 |
| LRQ (Ours) | 8/8/8 | 72.84 | 77.37 | 72.04 | 67.01 | 53.03 | 40.53 | 41.60 | **60.63** |
| Llama 13B | 16/16/16 | 68.53 | 79.11 | 76.23 | 70.01 | 59.89 | 44.54 | 42.20 | 62.93 |
| RTN | 8/8/16 | 66.06 | 71.82 | 65.70 | 62.98 | 50.97 | 35.58 | 36.60 | 55.67 |
| SmoothQuant | 8/8/16 | 68.29 | 75.30 | 71.82 | 68.03 | 55.18 | 40.19 | 41.20 | 60.00 |
| FlexRound | 8/8/16 | 68.59 | 78.67 | 75.21 | 70.64 | 58.88 | 43.60 | 41.20 | 62.40 |
| LRQ (Ours) | 8/8/16 | 68.99 | 79.22 | 75.61 | 71.19 | 58.92 | 43.52 | 43.00 | **62.92** |
| RTN | 8/8/8 | 65.87 | 72.25 | 62.52 | 62.19 | 51.81 | 35.41 | 38.40 | 55.49 |
| SmoothQuant | 8/8/8 | 67.34 | 75.19 | 71.78 | 69.06 | 54.92 | 40.44 | 38.80 | 59.65 |
| FlexRound | 8/8/8 | 68.78 | 78.51 | 75.23 | 70.56 | 58.46 | 44.03 | 41.00 | 62.37 |
| LRQ (Ours) | 8/8/8 | 68.84 | 78.78 | 75.56 | 70.80 | 59.13 | 44.62 | 41.60 | **62.76** |
| Llama 33B | 16/16/16 | 68.38 | 80.09 | 79.21 | 72.93 | 58.92 | 45.48 | 42.00 | 63.86 |
| RTN | 8/8/16 | 69.02 | 76.01 | 69.11 | 66.54 | 57.07 | 41.64 | 41.40 | 60.11 |
| SmoothQuant | 8/8/16 | 71.04 | 75.24 | 71.01 | 69.38 | 54.38 | 43.34 | 40.60 | 60.71 |
| FlexRound | 8/8/16 | 69.08 | 79.16 | 77.43 | 72.53 | 56.61 | 44.97 | 44.00 | 63.40 |
| LRQ (Ours) | 8/8/16 | 68.44 | 80.03 | 78.37 | 74.19 | 58.16 | 46.33 | 42.20 | **63.96** |
| RTN | 8/8/8 | 68.81 | 76.55 | 68.76 | 66.06 | 56.48 | 42.49 | 42.40 | 60.22 |
| SmoothQuant | 8/8/8 | 71.31 | 75.30 | 71.29 | 68.98 | 53.66 | 43.26 | 41.00 | 60.69 |
| FlexRound | 8/8/8 | 69.05 | 79.49 | 77.49 | 70.88 | 56.86 | 43.60 | 42.00 | 62.77 |
| LRQ (Ours) | 8/8/8 | 68.84 | 79.98 | 78.52 | 73.72 | 58.21 | 45.73 | 43.00 | **64.00** |
| Llama 65B | 16/16/16 | 82.32 | 80.85 | 80.71 | 77.19 | 58.71 | 46.33 | 44.60 | 67.24 |
| RTN | 8/8/16 | 79.48 | 77.04 | 74.15 | 71.19 | 52.48 | 43.52 | 43.80 | 63.09 |
| SmoothQuant | 8/8/16 | 78.72 | 78.84 | 79.12 | 74.03 | 56.23 | 45.22 | 43.20 | 65.05 |
| FlexRound | 8/8/16 | 81.31 | 79.33 | 79.16 | 73.56 | 57.83 | 46.08 | 44.60 | 65.98 |
| LRQ (Ours) | 8/8/16 | 82.45 | 80.69 | 79.92 | 76.64 | 58.92 | 46.67 | 45.60 | **67.27** |
| RTN | 8/8/8 | 79.51 | 75.79 | 74.13 | 71.35 | 51.85 | 44.03 | 43.60 | 62.89 |
| SmoothQuant | 8/8/8 | 78.78 | 79.54 | 79.11 | 73.32 | 56.23 | 45.90 | 43.80 | 65.24 |
| FlexRound | 8/8/8 | 80.46 | 79.38 | 79.23 | 74.98 | 57.20 | 46.42 | 45.00 | 66.10 |
| LRQ (Ours) | 8/8/8 | 82.35 | 81.12 | 79.96 | 75.61 | 58.96 | 46.59 | 45.40 | **67.14** |

Table 6: Five-shot performance of Llama on Massive Multitask Language Understanding with per-channel asymmetric weight quantization, per-tensor asymmetric static activation quantization, and per-token asymmetric KV cache quantization (if applied). Please refer to Figure 4. The accuracy (%) is reported for four groups of disciplines (STEM, Humanities, Social Science, and Other). The number of bits used for weights, activations, and KV cache is expressed as W/A/KV.

| Method | # Bits (W/A/KV) | STEM | Humanities | Social Science | Other | Average |
|---|---|---|---|---|---|---|
| Llama 7B | 16/16/16 | 30.58 | 33.88 | 38.19 | 38.25 | 35.12 |
| RTN | 8/8/16 | 27.40 | 27.16 | 29.18 | 30.38 | 28.40 |
| SmoothQuant | 8/8/16 | 28.36 | 27.89 | 32.63 | 30.41 | 29.61 |
| FlexRound | 8/8/16 | 28.30 | 29.20 | 30.13 | 33.47 | 30.20 |
| LRQ (Ours) | 8/8/16 | 29.69 | 32.48 | 37.63 | 38.80 | **34.47** |
| RTN | 8/8/8 | 27.04 | 27.23 | 29.28 | 30.38 | 28.36 |
| SmoothQuant | 8/8/8 | 28.40 | 28.69 | 32.79 | 30.48 | 29.94 |
| FlexRound | 8/8/8 | 27.60 | 28.71 | 29.61 | 31.99 | 29.43 |
| LRQ (Ours) | 8/8/8 | 29.72 | 32.79 | 37.44 | 38.16 | **34.39** |
| Llama 13B | 16/16/16 | 36.35 | 44.97 | 54.14 | 53.15 | 47.02 |
| RTN | 8/8/16 | 26.61 | 25.53 | 27.40 | 24.52 | 25.94 |
| SmoothQuant | 8/8/16 | 27.80 | 29.31 | 31.04 | 30.88 | 29.73 |
| FlexRound | 8/8/16 | 35.06 | 41.68 | 49.37 | 49.81 | 43.82 |
| LRQ (Ours) | 8/8/16 | 34.72 | 44.65 | 51.71 | 52.28 | **45.83** |
| RTN | 8/8/8 | 26.38 | 25.33 | 27.95 | 24.83 | 26.01 |
| SmoothQuant | 8/8/8 | 27.24 | 30.12 | 30.58 | 31.31 | 29.87 |
| FlexRound | 8/8/8 | 33.63 | 42.81 | 48.65 | 49.26 | 43.60 |
| LRQ (Ours) | 8/8/8 | 35.16 | 44.55 | 51.74 | 52.04 | **45.83** |
| Llama 33B | 16/16/16 | 46.69 | 56.39 | 67.40 | 63.60 | 58.38 |
| RTN | 8/8/16 | 32.14 | 32.22 | 37.11 | 38.25 | 34.67 |
| SmoothQuant | 8/8/16 | 38.17 | 41.45 | 50.37 | 51.08 | 44.92 |
| FlexRound | 8/8/16 | 43.94 | 52.31 | 62.14 | 60.21 | 54.49 |
| LRQ (Ours) | 8/8/16 | 45.13 | 52.99 | 64.12 | 61.88 | **55.79** |
| RTN | 8/8/8 | 32.47 | 32.37 | 38.35 | 40.59 | 35.60 |
| SmoothQuant | 8/8/8 | 37.94 | 41.64 | 50.57 | 51.48 | 45.07 |
| FlexRound | 8/8/8 | 43.47 | 52.20 | 61.94 | 59.90 | 54.24 |
| LRQ (Ours) | 8/8/8 | 45.26 | 52.58 | 63.99 | 61.26 | **55.51** |
| Llama 65B | 16/16/16 | 51.95 | 61.87 | 73.32 | 67.58 | 63.57 |
| RTN | 8/8/16 | 42.25 | 46.74 | 61.13 | 54.57 | 50.73 |
| SmoothQuant | 8/8/16 | 44.70 | 50.54 | 63.99 | 57.28 | 53.79 |
| FlexRound | 8/8/16 | 46.52 | 54.30 | 66.36 | 60.83 | 56.78 |
| LRQ (Ours) | 8/8/16 | 50.89 | 61.15 | 72.64 | 66.04 | **62.59** |
| RTN | 8/8/8 | 41.22 | 47.23 | 61.39 | 54.69 | 50.76 |
| SmoothQuant | 8/8/8 | 44.83 | 50.82 | 63.34 | 57.09 | 53.72 |
| FlexRound | 8/8/8 | 46.32 | 54.60 | 65.06 | 62.49 | 56.94 |
| LRQ (Ours) | 8/8/8 | 50.96 | 61.28 | 71.99 | 66.66 | **62.65** |

Table 7: Zero-shot performance of Llama 2 on common sense reasoning tasks (BoolQ, PIQA, HellaSwag, WinoGrande, ARC easy and challenge, and OpenBookQA) with per-channel asymmetric weight quantization, per-tensor asymmetric static activation quantization, and per-token asymmetric KV cache quantization (if applied). Please refer to Figure 4. The accuracy (%) is reported for common sense reasoning tasks. The number of bits used for weights, activations, and KV cache is expressed as W/A/KV.

| Method | # Bits (W/A/KV) | BoolQ | PIQA | HellaSwag | WinoGrande | ARC-e | ARC-c | OBQA | Average |
|---|---|---|---|---|---|---|---|---|---|
| Llama 2 7B | 16/16/16 | 71.07 | 76.99 | 72.96 | 67.25 | 53.58 | 40.53 | 40.80 | 60.45 |
| RTN | 8/8/16 | 60.86 | 67.19 | 57.53 | 59.43 | 45.50 | 32.00 | 34.20 | 50.96 |
| SmoothQuant | 8/8/16 | 67.06 | 72.91 | 67.24 | 64.72 | 50.72 | 35.84 | 38.20 | 56.67 |
| FlexRound | 8/8/16 | 71.99 | 77.04 | 71.23 | 65.11 | 54.42 | 40.44 | 38.80 | 59.86 |
| LRQ (Ours) | 8/8/16 | 67.49 | 77.58 | 72.19 | 67.96 | 54.76 | 39.59 | 40.40 | **60.00** |
| RTN | 8/8/8 | 60.58 | 67.08 | 57.66 | 60.54 | 45.83 | 31.57 | 34.40 | 51.09 |
| SmoothQuant | 8/8/8 | 67.09 | 73.29 | 67.26 | 64.25 | 51.18 | 36.26 | 38.40 | 56.82 |
| FlexRound | 8/8/8 | 72.05 | 77.26 | 71.30 | 65.98 | 54.88 | 39.16 | 39.20 | **59.98** |
| LRQ (Ours) | 8/8/8 | 67.86 | 76.99 | 71.97 | 67.01 | 54.71 | 40.19 | 40.00 | 59.82 |
| Llama 2 13B | 16/16/16 | 69.02 | 79.05 | 76.62 | 69.61 | 57.95 | 44.28 | 42.00 | 62.65 |
| RTN | 8/8/16 | 63.12 | 73.99 | 62.60 | 58.80 | 52.15 | 36.26 | 36.40 | 54.76 |
| SmoothQuant | 8/8/16 | 63.61 | 75.35 | 70.67 | 63.54 | 54.42 | 40.53 | 39.20 | 58.19 |
| FlexRound | 8/8/16 | 66.70 | 78.56 | 75.63 | 69.06 | 58.33 | 43.26 | 40.00 | 61.65 |
| LRQ (Ours) | 8/8/16 | 68.65 | 78.45 | 75.79 | 71.74 | 59.34 | 43.94 | 41.40 | **62.76** |
| RTN | 8/8/8 | 62.97 | 73.72 | 62.60 | 57.77 | 52.86 | 36.77 | 37.00 | 54.81 |
| SmoothQuant | 8/8/8 | 63.94 | 74.97 | 70.50 | 65.43 | 54.88 | 40.78 | 38.60 | 58.44 |
| FlexRound | 8/8/8 | 66.94 | 79.00 | 75.32 | 69.38 | 58.54 | 42.92 | 40.40 | 61.79 |
| LRQ (Ours) | 8/8/8 | 68.59 | 78.67 | 75.83 | 70.64 | 58.16 | 43.34 | 39.80 | **62.15** |
| Llama 2 70B | 16/16/16 | 76.70 | 80.85 | 80.85 | 76.95 | 59.72 | 47.95 | 44.40 | 66.77 |
| RTN | 8/8/16 | 73.27 | 78.18 | 76.89 | 69.69 | 57.91 | 45.90 | 41.60 | 63.35 |
| SmoothQuant | 8/8/16 | 77.13 | 76.99 | 79.46 | 71.90 | 55.39 | 45.14 | 43.60 | 64.23 |
| FlexRound | 8/8/16 | 75.81 | 80.25 | 79.03 | 74.59 | 59.43 | 46.42 | 43.40 | 65.56 |
| LRQ (Ours) | 8/8/16 | 77.71 | 80.69 | 79.83 | 74.11 | 57.91 | 45.99 | 43.60 | **65.69** |
| RTN | 8/8/8 | 72.39 | 78.51 | 76.49 | 69.61 | 57.74 | 44.62 | 40.40 | 62.82 |
| SmoothQuant | 8/8/8 | 76.73 | 77.15 | 79.37 | 73.01 | 55.56 | 46.59 | 43.20 | 64.52 |
| FlexRound | 8/8/8 | 76.18 | 80.36 | 79.09 | 75.06 | 60.10 | 46.42 | 43.80 | 65.86 |
| LRQ (Ours) | 8/8/8 | 77.95 | 81.23 | 79.78 | 74.82 | 57.83 | 46.33 | 43.60 | **65.93** |

Table 8: Five-shot performance of Llama 2 on Massive Multitask Language Understanding with per-channel asymmetric weight quantization, per-tensor asymmetric static activation quantization, and per-token asymmetric KV cache quantization (if applied). Please refer to Figure 4. The accuracy (%) is reported for four groups of disciplines (STEM, Humanities, Social Science, and Other). The number of bits used for weights, activations, and KV cache is expressed as W/A/KV.

| Method | # Bits (W/A/KV) | STEM | Humanities | Social Science | Other | Average |
|---|---|---|---|---|---|---|
| Llama 2 7B | 16/16/16 | 37.04 | 43.38 | 51.84 | 52.44 | 45.96 |
| RTN | 8/8/16 | 28.26 | 24.65 | 31.39 | 24.68 | 26.91 |
| SmoothQuant | 8/8/16 | 21.97 | 24.51 | 22.00 | 24.28 | 23.36 |
| FlexRound | 8/8/16 | 32.70 | 38.38 | 43.58 | 45.77 | 40.01 |
| LRQ (Ours) | 8/8/16 | 34.36 | 40.02 | 46.64 | 47.32 | **41.94** |
| RTN | 8/8/8 | 29.66 | 24.06 | 30.45 | 24.49 | 26.76 |
| SmoothQuant | 8/8/8 | 21.67 | 25.06 | 22.26 | 24.03 | 23.48 |
| FlexRound | 8/8/8 | 33.40 | 36.96 | 43.13 | 46.30 | 39.70 |
| LRQ (Ours) | 8/8/8 | 34.82 | 39.91 | 46.47 | 47.62 | **42.04** |
| Llama 2 13B | 16/16/16 | 44.27 | 54.43 | 63.41 | 60.76 | 55.68 |
| RTN | 8/8/16 | 29.16 | 24.38 | 30.52 | 29.49 | 27.93 |
| SmoothQuant | 8/8/16 | 21.24 | 24.29 | 21.71 | 23.84 | 22.97 |
| FlexRound | 8/8/16 | 41.95 | 51.20 | 60.90 | 59.65 | 53.29 |
| LRQ (Ours) | 8/8/16 | 41.09 | 51.58 | 61.39 | 59.41 | **53.28** |
| RTN | 8/8/8 | 29.06 | 24.23 | 29.93 | 29.03 | 27.62 |
| SmoothQuant | 8/8/8 | 21.31 | 24.08 | 21.71 | 23.72 | 22.88 |
| FlexRound | 8/8/8 | 41.09 | 51.58 | 61.39 | 59.41 | 53.28 |
| LRQ (Ours) | 8/8/8 | 42.88 | 51.97 | 62.14 | 59.93 | **54.08** |
| Llama 2 70B | 16/16/16 | 57.79 | 65.16 | 80.44 | 74.61 | 69.11 |
| RTN | 8/8/16 | 45.99 | 52.69 | 65.52 | 59.16 | 55.58 |
| SmoothQuant | 8/8/16 | 48.31 | 54.35 | 69.94 | 63.05 | 58.47 |
| FlexRound | 8/8/16 | 53.64 | 61.36 | 77.35 | 71.90 | 65.64 |
| LRQ (Ours) | 8/8/16 | 54.41 | 62.78 | 77.48 | 71.56 | **66.23** |
| RTN | 8/8/8 | 46.82 | 53.37 | 66.23 | 58.51 | 55.97 |
| SmoothQuant | 8/8/8 | 47.28 | 54.60 | 69.32 | 63.33 | 58.27 |
| FlexRound | 8/8/8 | 54.27 | 61.11 | 77.45 | 71.31 | 65.57 |
| LRQ (Ours) | 8/8/8 | 54.44 | 62.61 | 76.99 | 71.78 | **66.12** |

Table 9: Zero-shot performance of Llama 2 on common sense reasoning tasks (BoolQ, PIQA, HellaSwag, WinoGrande, ARC easy and challenge, and OpenBookQA) with per-channel asymmetric weight quantization, per-token asymmetric activation quantization, and per-token asymmetric KV cache quantization (if applied). Please refer to Figure 6. The accuracy (%) is reported for common sense reasoning tasks. The number of bits used for weights, activations, and KV cache is expressed as W/A/KV.

| Method | # Bits (W/A/KV) | BoolQ | PIQA | HellaSwag | WinoGrande | ARC-e | ARC-c | OBQA | Average |
|---|---|---|---|---|---|---|---|---|---|
| Llama 2 7B | 16/16/16 | 71.07 | 76.99 | 72.96 | 67.25 | 53.58 | 40.53 | 40.80 | 60.45 |
| RTN | 8/8/16 | 69.54 | 76.93 | 72.21 | 67.17 | 53.24 | 41.04 | 40.60 | 60.10 |
| SmoothQuant | 8/8/16 | 70.15 | 77.04 | 72.91 | 67.01 | 53.62 | 40.70 | 41.00 | 60.35 |
| FlexRound | 8/8/16 | 72.26 | 76.88 | 72.57 | 66.93 | 53.70 | 40.36 | 40.40 | 60.44 |
| LRQ (Ours) | 8/8/16 | 72.54 | 77.15 | 72.58 | 67.09 | 53.70 | 41.04 | 40.40 | **60.64** |
| RTN | 8/8/8 | 69.60 | 77.20 | 72.26 | 67.09 | 53.62 | 39.85 | 41.00 | 60.09 |
| SmoothQuant | 8/8/8 | 70.73 | 76.77 | 72.80 | 67.25 | 53.70 | 41.04 | 40.60 | 60.41 |
| FlexRound | 8/8/8 | 72.02 | 77.09 | 72.50 | 67.40 | 54.17 | 40.19 | 40.80 | 60.60 |
| LRQ (Ours) | 8/8/8 | 72.45 | 77.04 | 72.70 | 67.09 | 53.66 | 40.61 | 41.60 | **60.74** |
| RTN | 4/8/16 | 67.95 | 74.32 | 65.84 | 62.12 | 46.68 | 37.20 | 35.80 | 55.70 |
| SmoothQuant | 4/8/16 | 53.85 | 69.26 | 51.34 | 55.96 | 40.70 | 31.66 | 35.20 | 48.28 |
| FlexRound | 4/8/16 | 71.96 | 77.04 | 72.17 | 65.59 | 53.58 | 39.85 | 40.20 | 60.06 |
| LRQ (Ours) | 4/8/16 | 72.94 | 76.88 | 71.85 | 65.27 | 53.96 | 39.85 | 40.80 | **60.22** |
| RTN | 4/8/8 | 68.13 | 75.14 | 65.89 | 62.67 | 46.42 | 36.52 | 36.20 | 55.85 |
| SmoothQuant | 4/8/8 | 53.27 | 69.64 | 51.28 | 56.35 | 40.95 | 31.83 | 35.00 | 48.33 |
| FlexRound | 4/8/8 | 71.71 | 76.77 | 72.24 | 66.14 | 53.49 | 40.02 | 40.40 | 60.11 |
| LRQ (Ours) | 4/8/8 | 73.00 | 76.99 | 71.90 | 65.98 | 54.38 | 39.68 | 41.20 | **60.45** |
| Llama 2 13B | 16/16/16 | 69.02 | 79.05 | 76.62 | 69.61 | 57.95 | 44.28 | 42.00 | 62.65 |
| RTN | 8/8/16 | 67.80 | 78.89 | 75.61 | 68.90 | 58.08 | 43.69 | 41.60 | 62.08 |
| SmoothQuant | 8/8/16 | 69.36 | 79.33 | 76.47 | 69.69 | 58.04 | 44.54 | 42.40 | **62.83** |
| FlexRound | 8/8/16 | 69.27 | 78.73 | 76.62 | 69.69 | 57.62 | 44.71 | 42.20 | 62.69 |
| LRQ (Ours) | 8/8/16 | 69.24 | 78.67 | 76.48 | 69.30 | 57.79 | 44.03 | 42.40 | 62.56 |
| RTN | 8/8/8 | 67.46 | 78.73 | 75.57 | 68.51 | 58.12 | 44.28 | 41.40 | 62.01 |
| SmoothQuant | 8/8/8 | 69.33 | 79.05 | 76.52 | 69.38 | 58.04 | 44.37 | 42.00 | 62.67 |
| FlexRound | 8/8/8 | 69.36 | 79.16 | 76.67 | 69.53 | 57.83 | 44.37 | 42.80 | **62.82** |
| LRQ (Ours) | 8/8/8 | 69.02 | 78.78 | 76.48 | 69.93 | 57.83 | 43.86 | 42.00 | 62.56 |
| RTN | 4/8/16 | 65.20 | 73.61 | 60.00 | 58.80 | 49.12 | 36.18 | 34.80 | 53.96 |
| SmoothQuant | 4/8/16 | 62.26 | 61.21 | 44.59 | 53.67 | 36.95 | 27.99 | 32.80 | 45.64 |
| FlexRound | 4/8/16 | 69.14 | 78.67 | 75.67 | 68.98 | 58.92 | 44.20 | 41.00 | 62.37 |
| LRQ (Ours) | 4/8/16 | 71.10 | 78.29 | 75.75 | 69.30 | 57.74 | 43.69 | 41.00 | **62.41** |
| RTN | 4/8/8 | 65.23 | 74.05 | 60.04 | 58.64 | 49.07 | 35.92 | 35.20 | 54.02 |
| SmoothQuant | 4/8/8 | 62.17 | 61.37 | 44.59 | 53.67 | 36.99 | 28.24 | 32.40 | 45.63 |
| FlexRound | 4/8/8 | 69.05 | 78.51 | 75.51 | 69.53 | 58.75 | 43.60 | 41.20 | 62.31 |
| LRQ (Ours) | 4/8/8 | 71.13 | 78.29 | 75.79 | 68.90 | 57.83 | 43.34 | 41.20 | **62.35** |
| Llama 2 70B | 16/16/16 | 76.70 | 80.85 | 80.85 | 76.95 | 59.72 | 47.95 | 44.40 | 66.77 |
| RTN | 8/8/16 | 76.02 | 81.07 | 80.37 | 76.01 | 60.14 | 48.04 | 44.40 | 66.58 |
| SmoothQuant | 8/8/16 | 76.21 | 81.12 | 80.72 | 76.40 | 59.39 | 47.53 | 44.80 | 66.60 |
| FlexRound | 8/8/16 | 75.72 | 81.56 | 80.60 | 75.77 | 60.19 | 48.89 | 44.80 | **66.79** |
| LRQ (Ours) | 8/8/16 | 75.84 | 81.66 | 80.64 | 75.93 | 60.40 | 48.38 | 44.00 | 66.69 |
| RTN | 8/8/8 | 76.02 | 81.07 | 80.45 | 75.61 | 60.31 | 47.87 | 43.80 | 66.45 |
| SmoothQuant | 8/8/8 | 76.15 | 80.96 | 80.63 | 77.11 | 59.09 | 47.87 | 44.60 | 66.63 |
| FlexRound | 8/8/8 | 75.93 | 81.45 | 80.48 | 75.85 | 60.06 | 48.55 | 44.80 | 66.73 |
| LRQ (Ours) | 8/8/8 | 75.99 | 81.50 | 80.61 | 75.77 | 59.97 | 49.49 | 45.20 | **66.93** |
| RTN | 4/8/16 | 75.63 | 78.73 | 71.28 | 69.61 | 53.24 | 43.34 | 40.20 | 61.72 |
| SmoothQuant | 4/8/16 | 66.12 | 75.79 | 56.06 | 60.54 | 50.38 | 35.67 | 39.60 | 54.88 |
| FlexRound | 4/8/16 | 77.80 | 80.90 | 80.06 | 74.66 | 60.31 | 47.61 | 43.60 | 66.42 |
| LRQ (Ours) | 4/8/16 | 77.92 | 80.74 | 80.38 | 75.14 | 60.35 | 47.95 | 42.80 | **66.47** |
| RTN | 4/8/8 | 75.90 | 79.22 | 71.39 | 70.56 | 53.11 | 43.60 | 40.40 | 62.03 |
| SmoothQuant | 4/8/8 | 66.24 | 75.84 | 56.19 | 60.46 | 50.25 | 36.01 | 40.40 | 55.06 |
| FlexRound | 4/8/8 | 77.31 | 80.96 | 79.89 | 75.30 | 60.19 | 48.21 | 43.40 | 66.47 |
| LRQ (Ours) | 4/8/8 | 77.92 | 81.28 | 80.42 | 75.06 | 60.94 | 48.04 | 42.60 | **66.61** |

Table 10: Five-shot performance of Llama 2 on Massive Multitask Language Understanding with per-channel asymmetric weight quantization, per-token asymmetric activation quantization, and per-token asymmetric KV cache quantization (if applied). Please refer to Figure 6. The accuracy (%) is reported for four groups of disciplines (STEM, Humanities, Social Science, and Other). The number of bits used for weights, activations, and KV cache is expressed as W/A/KV.

| Method | # Bits (W/A/KV) | STEM | Humanities | Social Science | Other | Average |
|---|---|---|---|---|---|---|
| Llama 2 7B | 16/16/16 | 37.04 | 43.38 | 51.84 | 52.44 | 45.96 |
| RTN | 8/8/16 | 36.41 | 42.49 | 50.31 | 52.47 | 45.20 |
| SmoothQuant | 8/8/16 | 37.28 | 43.00 | 52.13 | 52.65 | 46.00 |
| FlexRound | 8/8/16 | 36.38 | 42.91 | 51.80 | 52.87 | 45.76 |
| LRQ (Ours) | 8/8/16 | 36.91 | 43.27 | 52.19 | 52.78 | **46.05** |
| RTN | 8/8/8 | 36.15 | 42.85 | 50.34 | 52.31 | 45.24 |
| SmoothQuant | 8/8/8 | 36.98 | 42.93 | 51.87 | 52.56 | 45.83 |
| FlexRound | 8/8/8 | 36.98 | 42.91 | 51.87 | 52.28 | 45.76 |
| LRQ (Ours) | 8/8/8 | 36.88 | 43.12 | 51.67 | 52.53 | **45.83** |
| RTN | 4/8/16 | 27.63 | 25.87 | 27.82 | 28.32 | 27.24 |
| SmoothQuant | 4/8/16 | 26.01 | 24.80 | 22.16 | 26.71 | 24.93 |
| FlexRound | 4/8/16 | 37.01 | 42.40 | 50.80 | 50.34 | 44.92 |
| LRQ (Ours) | 4/8/16 | 36.78 | 42.66 | 51.19 | 51.73 | **45.36** |
| RTN | 4/8/8 | 28.00 | 25.80 | 27.53 | 28.01 | 27.16 |
| SmoothQuant | 4/8/8 | 25.75 | 24.91 | 22.49 | 26.59 | 24.95 |
| FlexRound | 4/8/8 | 37.81 | 42.55 | 50.47 | 50.65 | 45.14 |
| LRQ (Ours) | 4/8/8 | 36.88 | 42.53 | 50.80 | 52.22 | **45.36** |
| Llama 2 13B | 16/16/16 | 44.27 | 54.43 | 63.41 | 60.76 | 55.68 |
| RTN | 8/8/16 | 43.57 | 52.88 | 61.88 | 61.17 | 54.76 |
| SmoothQuant | 8/8/16 | 43.67 | 53.39 | 63.60 | 60.76 | 55.24 |
| FlexRound | 8/8/16 | 43.84 | 53.65 | 63.37 | 61.10 | 55.39 |
| LRQ (Ours) | 8/8/16 | 44.80 | 53.75 | 63.47 | 60.73 | **55.57** |
| RTN | 8/8/8 | 43.87 | 52.88 | 62.33 | 60.67 | 54.81 |
| SmoothQuant | 8/8/8 | 43.74 | 53.20 | 63.18 | 60.83 | 55.11 |
| FlexRound | 8/8/8 | 44.17 | 52.88 | 63.76 | 61.29 | 55.33 |
| LRQ (Ours) | 8/8/8 | 44.50 | 53.07 | 63.24 | 61.26 | **55.35** |
| RTN | 4/8/16 | 30.55 | 26.08 | 33.51 | 35.07 | 30.74 |
| SmoothQuant | 4/8/16 | 28.20 | 25.08 | 27.07 | 27.64 | 26.78 |
| FlexRound | 4/8/16 | 42.91 | 50.80 | 62.11 | 60.27 | 53.77 |
| LRQ (Ours) | 4/8/16 | 43.24 | 52.41 | 61.78 | 60.24 | **54.30** |
| RTN | 4/8/8 | 30.95 | 26.31 | 32.92 | 34.58 | 30.67 |
| SmoothQuant | 4/8/8 | 27.87 | 24.95 | 26.58 | 27.91 | 26.62 |
| FlexRound | 4/8/8 | 42.88 | 50.71 | 61.94 | 59.93 | 53.77 |
| LRQ (Ours) | 4/8/8 | 43.90 | 52.56 | 62.07 | 59.96 | **54.49** |
| Llama 2 70B | 16/16/16 | 57.79 | 65.16 | 80.44 | 74.61 | 69.11 |
| RTN | 8/8/16 | 56.06 | 63.00 | 78.32 | 73.10 | 67.20 |
| SmoothQuant | 8/8/16 | 57.59 | 64.40 | 80.40 | 74.15 | 68.69 |
| FlexRound | 8/8/16 | 57.69 | 63.80 | 79.98 | 73.63 | 68.30 |
| LRQ (Ours) | 8/8/16 | 57.95 | 64.48 | 80.21 | 73.90 | **68.70** |
| RTN | 8/8/8 | 56.23 | 63.55 | 78.39 | 73.01 | 67.41 |
| SmoothQuant | 8/8/8 | 57.59 | 64.21 | 80.70 | 74.58 | **68.79** |
| FlexRound | 8/8/8 | 57.22 | 63.97 | 79.62 | 73.81 | 68.22 |
| LRQ (Ours) | 8/8/8 | 57.95 | 63.85 | 80.34 | 73.94 | 68.52 |
| RTN | 4/8/16 | 41.12 | 45.72 | 56.78 | 53.49 | 48.95 |
| SmoothQuant | 4/8/16 | 29.69 | 30.61 | 36.50 | 37.60 | 33.31 |
| FlexRound | 4/8/16 | 59.96 | 62.98 | 79.04 | 73.23 | 67.56 |
| LRQ (Ours) | 4/8/16 | 56.46 | 64.59 | 79.07 | 72.83 | **67.92** |
| RTN | 4/8/8 | 41.19 | 45.74 | 57.52 | 53.61 | 49.16 |
| SmoothQuant | 4/8/8 | 29.69 | 31.31 | 36.89 | 37.42 | 33.59 |
| FlexRound | 4/8/8 | 56.26 | 62.89 | 78.78 | 72.92 | 67.26 |
| LRQ (Ours) | 4/8/8 | 55.57 | 64.65 | 78.97 | 72.52 | **67.65** |

## C  IMPLEMENTATION DETAILS

Table 11: Learning rate and batch size for FlexRound and LRQ when employing a per-tensor asymmetric static activation quantization scheme (see Figure 4) in Table 1, 2, 5, 6, 7, and 8.

| Method | Configuration | Llama 7B | Llama 13B | Llama 33B | Llama 65B | Llama 2 7B | Llama 2 13B | Llama 2 70B |
|--------|---------------|----------|-----------|-----------|-----------|------------|-------------|-------------|
| FlexRound | Learning rate | 3e-3 | 3e-3 | 1e-3 | 2e-3 | 3e-3 | 3e-3 | 1e-3 |
|           | Batch size    | 4    | 4    | 2    | 2    | 2    | 2    | 2    |
| LRQ       | Learning rate | 3e-3 | 2e-3 | 1.5e-3 | 1e-3 | 1e-3 | 1.5e-3 | 1e-3 |
|           | Batch size    | 2    | 2    | 2    | 2    | 2    | 2    | 2    |

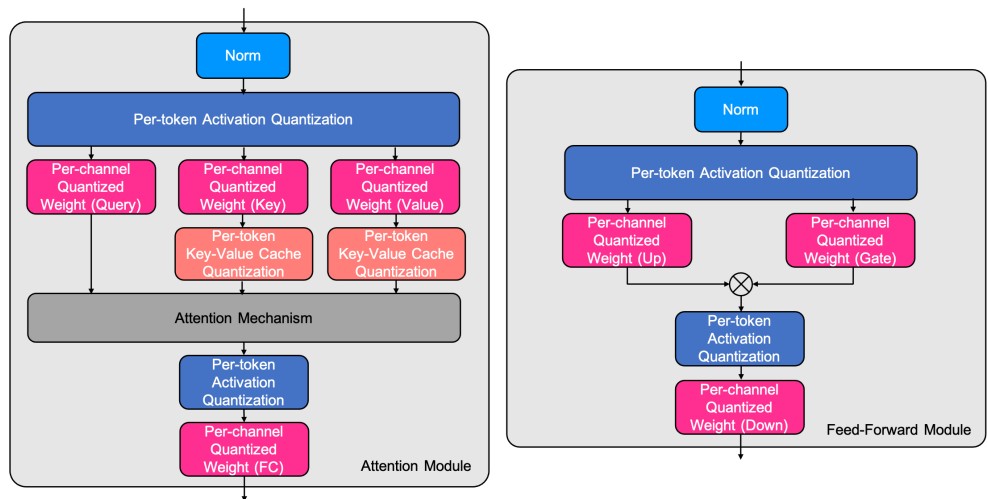

Figure 6: Illustration of a quantized Transformer block with per-channel asymmetric weight quantization, *per-token* asymmetric activation quantization, and per-token asymmetric KV cache quantization. We remain the inputs of softmax and normalization layers in FP16.

Table 12: Learning rate for FlexRound and LRQ when adopting a per-token asymmetric activation quantization scheme (see Figure 6) in Table 3, 4, 9, and 10.

| Method | Weight | Llama 2 7B | Llama 2 13B | Llama 2 70B |
|--------|--------|------------|-------------|-------------|
| FlexRound | 8-bit | 1e-4 | 4e-4 | 3e-4 |
|           | 4-bit | 5e-4 | 4e-4 | 5e-4 |
| LRQ       | 8-bit | 1e-4 | 2e-4 | 4e-4 |
|           | 4-bit | 5e-4 | 5e-4 | 4e-4 |

For the quantization scheme depicted in Figure 4, both FlexRound and LRQ are implemented in the experimental setting of QDrop (Wei et al., 2022) with the exception of the number of iterations for block-wise reconstruction, the batch size, and the learning rate. For all the Llama and Llama 2 models, the number of iterations for block-wise reconstruction is set to 5000 for both FlexRound and LRQ. The learning rate and the batch size for FlexRound and LRQ are described in 11. Notice that when applying LRQ to Llama 2 70B, the key and value projection weights are quantized via not LRQ but FlexRound due to the presence of GQA (Ainslie et al., 2023) in Llama 2 70B. To obtain the experimental results in Table 1 and 2, per-token asymmetric KV cache quantization is applied after completing block-wise reconstruction for all the Transformer blocks.

In the case of quantization scheme indicated in Figure 6, both FlexRound and LRQ are first implemented in the experimental setting of BRECQ (Li et al., 2021) with the exception of the number of

iterations for block-wise reconstruction, the batch size, and the learning rate. The number of iterations for block-wise reconstruction and the batch size are set to $5000$ and $2$ respectively, for every Llama 2 model regardless of the number of bits used for weights. Table 12 exhibits the learning rate for FlexRound and LRQ in the case of 8-bit and 4-bit weight quantization, respectively. As explained in the above paragraph, when LRQ is applied to Llama 2 70B, weights in key and value projections are quantized via FlexRound. Here, when quantizing Llama 2 7B into $4$-bit via LRQ, the attention module is quantized via LRQ, but the feed-forward module is quantized via FlexRound. In addition, when quantizing Llama 2 70B into $4$-bit via LRQ, the feed-forward module is quantized via LRQ, but the attention module is quantized via FlexRound. To gain the experimental results in Table 3 and 4, per-token asymmetric activation quantization and per-token asymmetric KV cache quantization are sequentially applied after finishing block-wise reconstruction for all the Transformer blocks.

All experiments about SmoothQuant are conducted based on the code provided in the SmoothQuant github repository[1]. Following Xiao et al. (2022), we select $\alpha$, the hyperparameter to determine how much difficulty of activation quantization to shift to weight quantization, to be $0.8$ for both Llama and Llama 2 models.

For evaluation code, we utilize Eleuther AI's *lm-evaluation-harness* (Gao et al., 2021) for common sense reasoning tasks and follow the evaluation method in the MMLU github repository[2] for the MMLU benchmark.

---

[1] https://github.com/mit-han-lab/smoothquant
[2] https://github.com/hendrycks/test

## D   COMBINATION OF SMOOTHQUANT WITH FLEXROUND AND LRQ

Table 13: Zero-shot performance of Llama 7B on common sense reasoning tasks (BoolQ, PIQA, HellaSwag, WinoGrande, ARC easy and challenge, and OpenBookQA) with per-channel asymmetric weight quantization and per-tensor asymmetric static activation quantization, while keeping the KV cache in FP16. Here, 'SQ + FlexRound' and 'SQ + LRQ' denote FlexRound and LRQ that initially begin their own learning process from the SmoothQuant baseline in lieu of the rounding-to-nearest baseline, respectively. The accuracy (%) is reported for common sense reasoning tasks. The number of bits used for weights, activations, and KV cache is expressed as W/A/KV.

| Method | # Bits (W/A/KV) | BoolQ | PIQA | HellaSwag | WinoGrande | ARC-e | ARC-c | OBQA | Average |
|--------|-----------------|-------|------|-----------|------------|-------|-------|------|---------|
| Llama 7B | 16/16/16 | 73.15 | 77.31 | 72.96 | 67.09 | 52.48 | 41.38 | 42.40 | 60.97 |
| FlexRound | 8/8/16 | 73.76 | 76.66 | 71.75 | 67.01 | 52.31 | 40.02 | 42.20 | **60.53** |
| SQ+FlexRound | 8/8/16 | 73.85 | 76.77 | 71.54 | 66.38 | 51.43 | 40.44 | 41.60 | 60.29 |
| LRQ | 8/8/16 | 73.03 | 77.64 | 72.10 | 66.77 | 52.95 | 40.87 | 41.60 | **60.71** |
| SQ+LRQ | 8/8/16 | 73.15 | 76.88 | 72.24 | 66.38 | 52.86 | 40.61 | 40.60 | 60.39 |

Table 14: Five-shot performance of Llama 7B on Massive Multitask Language Understanding with per-channel asymmetric weight quantization and per-tensor asymmetric static activation quantization, while keeping the KV cache in FP16. Here, 'SQ + FlexRound' and 'SQ + LRQ' denote FlexRound and LRQ that initially begin their own learning process from the SmoothQuant baseline in lieu of the rounding-to-nearest baseline, respectively. The accuracy (%) is reported for four groups of disciplines (STEM, Humanities, Social Science, and Other). The number of bits used for weights, activations, and KV cache is expressed as W/A/KV.

| Method | # Bits (W/A/KV) | STEM | Humanities | Social Science | Other | Average |
|--------|-----------------|------|------------|----------------|-------|---------|
| Llama 7B | 16/16/16 | 30.58 | 33.88 | 38.19 | 38.25 | 35.12 |
| FlexRound | 8/8/16 | 28.30 | 29.20 | 30.13 | 33.47 | 30.20 |
| SQ+FlexRound | 8/8/16 | 30.98 | 29.71 | 33.80 | 35.26 | **32.16** |
| LRQ | 8/8/16 | 29.69 | 32.48 | 37.63 | 38.80 | **34.47** |
| SQ+LRQ | 8/8/16 | 30.35 | 31.84 | 37.44 | 37.32 | 34.01 |

As SmoothQuant is orthogonal to block-wise reconstruction, one might wonder how the performance of FlexRound and LRQ would change when FlexRound and LRQ start their own learning process from the SmoothQuant baseline in place of the RTN baseline. Table 13 and 14 reveal the performance of 'SmoothQuant (SQ) + FlexRound' and 'SmoothQuant (SQ) + LRQ' on common sense reasoning benchmarks and the MMLU benchmark, respectively. Unfortunately, in most cases, SmoothQuant does not display its efficacy when combined with FlexRound and LRQ. Although SmoothQuant enhances five-shot performance of FlexRound on MMLU by almost two percent, 'SQ + FlexRound' still underperforms LRQ as well as 'SQ + LRQ' on MMLU, which implies that employing low-rank weight-scaling matrices would be a better choice than using full weight-scaling matrices with additional pre-processing like an uniform per-channel scaling transformation in SmoothQuant.

# E    FIGURES OF ACCUMULATED RMSE ON ASSORTED SAMPLES

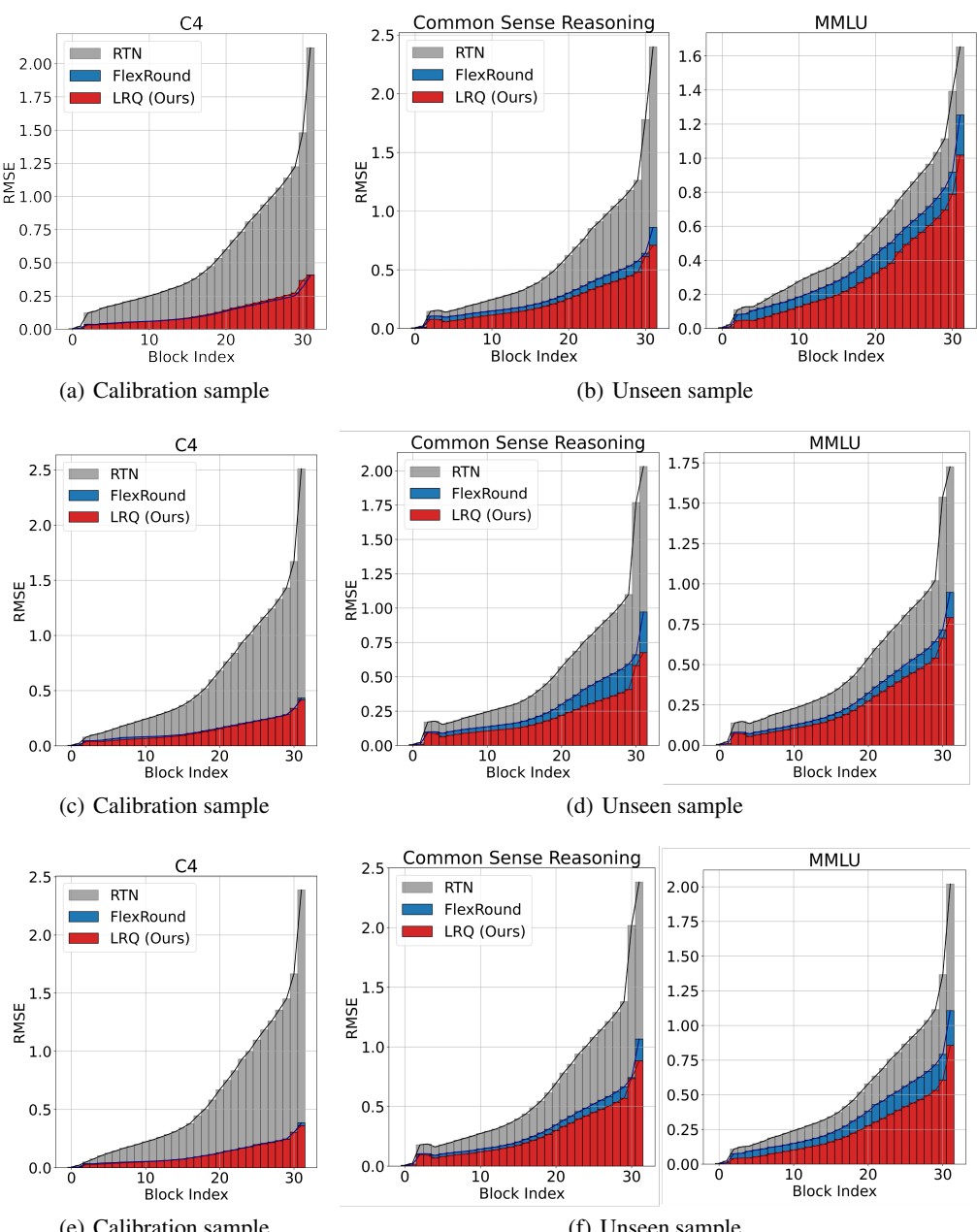

Figure 7: Accumulated root mean square error (RMSE) between $\boldsymbol{WX}$ and $\widehat{\boldsymbol{W}}\widetilde{\boldsymbol{X}}$ for RTN, FlexRound, and LRQ on (a), (c), (e) three different calibration samples from the C4 dataset and (b), (d), (f) three different unseen samples from common sense reasoning and MMLU benchmarks, ranging from the first Transformer block to the last Transformer block of Llama 7B. Here, weights and activations are quantized to 8-bit with per-channel asymmetric quantization and per-tensor asymmetric static quantization, respectively. Note that RMSE tends to rise in line with the block index due to the presence of $\widetilde{\boldsymbol{X}}$ that accumulates quantization error resulting from previous quantized Transformer blocks.

# F  SENSITIVITY OF ACCUMULATED RMSE TO THE NUMBER OF CALIBRATION SAMPLES

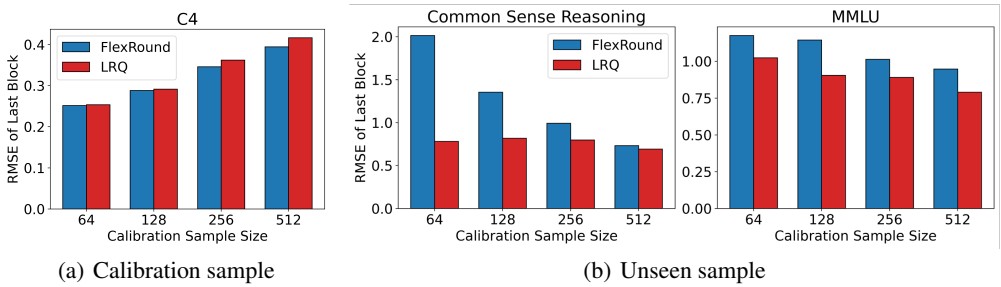

Figure 8: Accumulated root mean square error (RMSE) between $\boldsymbol{W}\boldsymbol{X}$ and $\widehat{\boldsymbol{W}}\widetilde{\boldsymbol{X}}$ for FlexRound and LRQ on (a) a calibration sample from the C4 dataset and (b) an unseen sample from common sense reasoning and MMLU benchmarks at the last Transformer block of Llama 7B. Here, weights and activations are quantized to 8-bit with per-channel asymmetric quantization and per-tensor asymmetric static quantization, respectively.

To figure out the sensitivity of accumulated root mean square error (RMSE) to the number of calibration samples used for the block-wise reconstruction, we compare accumulated RMSE between $\boldsymbol{W}\boldsymbol{X}$ and $\widehat{\boldsymbol{W}}\widetilde{\boldsymbol{X}}$ for FlexRound and LRQ at the last Transformer block of Llama 7B with the number of calibration samples varying from $64$ to $512$. As depicted in Figure 8(a), the accumulated RMSE of the last Transformer block on a calibration sample diminishes with a reduction in the number of calibration samples. This phenomenon is because FlexRound and LRQ are more likely to be fitted to calibration samples as the number of calibration samples becomes smaller. Conversely, Figure 8(b) reveals that the accumulated RMSE of the last Transformer block on each unseen sample from common sense reasoning and MMLU decreases with a larger number of calibration samples.

Notably, the pattern elucidated in Section 2.4 persists consistently across varying calibration sample sizes from $64$ to $512$. In other words, for every calibration sample size spanning from $64$ to $512$, LRQ consistently attains nearly identical accumulated RMSE to FlexRound for a calibration sample from the C4 dataset. Concurrently, the accumulated RMSE of LRQ remains markedly smaller than that of FlexRound for an unseen sample from common sense reasoning and MMLU. This observation provides additional support for the insight presented in Figure 3, as discussed in Section 2.4.

## G   EFFECT OF THE RANK $r$ ON THE ACCURACY OF LRQ

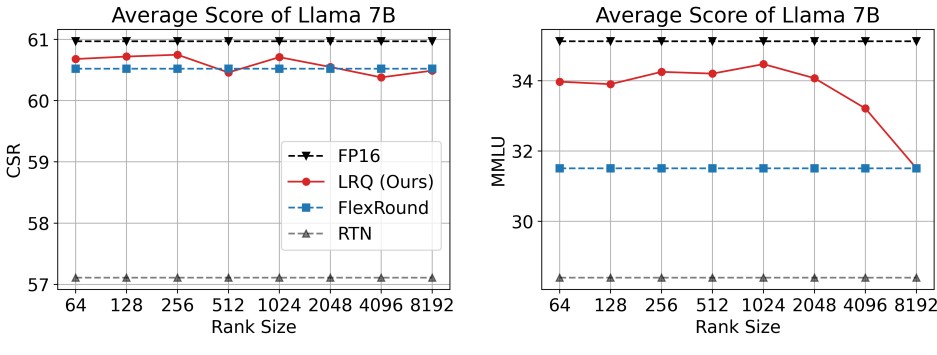

(a) Common Sense Reasoning tasks          (b) Massive Multitask Language Understanding

Figure 9: (a) Zero-shot performance and (b) five-shot performance of Llama 7B according to the rank $r$ in Eq. 2, spanning from $64$ to $8192$, where weights and activations are quantized to $8$-bit as described in Figure 4 while the KV cache is kept in FP16.

As outlined in Figure 9, the accuracy of LRQ either remains relatively stable (Figure 9(a)) or increases gradually (Figure 9(b)) with the rise in the rank $r$ from $64$ to $1024$. However, as the rank $r$ continuously increases from $2048$ to $8192$, the accuracy of LRQ eventually declines to match that of FlexRound on both common sense reasoning and MMLU. In light of these findings, it can be concluded that the employment of lower rank contributes considerably to improving the accuracy/generalization of quantized LLMs.

