# OpenReview forum: "LRQ: Optimizing Post-Training Quantization for Large Language Models by Learning Low-Rank Weight-Scaling Matrices"
_ICLR.cc/2024/Conference — Submitted to ICLR 2024_

### Official Review · Reviewer_VLPY · 2023-10-31

**Soundness:** 2 fair
**Presentation:** 2 fair
**Contribution:** 1 poor
**Rating:** 3
**Confidence:** 5

**Summary:**

The paper proposes a new weight rounding scheme via a small modification on top of FlexRound technique: whereas in the FlexRound the full matrix of elementwise reciprocals are being learned, the proposed method changes it to a low-rank matrix. In other words, if the FlexRound's quantization (rounding) function is given as $Q(\mathbf{W}; \alpha, \mathbf{S}) = \alpha \times \lfloor \frac{\mathbf{W}}{\mathbf S} \rceil$ where $\alpha$ is the shared re-scaling parameter, $\mathbf{S}$ is the matrix of learned reciprocals, and $\lfloor \cdot \rceil$ is the rounding-to-nearest function, then the proposed modification is to change the $\mathbf{S}$ to be of a low rank (with $r \in \{1024, 2048\}$ ) . The  parameters of this quantization function are learned in a block-wise reconstruction manner to minimize the $\ell_2$ difference between quantized and non-quantized outputs. Gradients for non-differentiable operations are computed via straight-through-estimation.
The motivation for the proposed change is rooted in the fact that learning the entire matrix $\mathbf{S}$ on the calibration dataset might lead to overfitting, and thus low-rank $\mathbf{S}$ will counteract it. The paper is supplemented with many experiments showcasing the effectiveness of the proposed method and comparisons to other techniques.

**Strengths:**

The proposed change requires minimal modification for any existing quantization pipelines and codebases, and thus can be quickly absorbed into actual real-world usage.
Despite simplicity, the experimental evaluation suggests that the method provides measurable, albeit marginal, improvement over existing techniques.

**Weaknesses:**

Despite the aforementioned strengths, I am not very convinced on why these changes are helping the quantization. The original motivation to introduce low-rank $\mathbf{S}$ was to combat overfitting via "decreasing the number of learnable parameters". However, in the provided ablation study between the rank of the matrix $\mathbf{S}$ and the achieved zere-shot accuracy (figure 5a) on a task, it seems the rank has minimal effect on the accuracy (red curve is almost flat). Then, the question would be what is contributing to the improved accuracy if not lower rank?

Improvements: whereas the presented results show indeed better accuracies, I am somewhat unsure to which degree these results showcase the best of the other methdos. For instance, it seems authors used same scale of 0.8 for SmoothQuant, whereas the scale should ideally be validated on a calibration data. Also, the same comparison to SmoothQuant seems to be unfair: smoothquant is pure no-learning post-training quantization technique, but FlexQuant and the suggested methods are actually introduces a learning objective that is optimized.  In many cases the difference between the suggested method and FlexQuant is such a minuscule number, that it can be attributed to a sampling noise (see table 1).

**Questions:**

1. Can you please address weaknesses raised above?
2. The FlexRound method as presented in the original paper does not have exponentiation on $mathbf{S}$, but the proposed method does have it. Is it a typo or the additional reparametrization on top of FlexRound?
3. What is the effect of $r$ and $c$ (eq.2)? Some ablation studies on those would be helpful
4. Some general recommendations on choosing rank, learning rates, and training regime would be of great help to all practitioners. Also, if you can tabularize the actual experiment details presented in appendix C; it would help to reproducibility as well
5. There are some minor typos in the paper: in section 2.1, we concentrate**s** => we concentrate in section 4, simeple => simple,

---

> ### Author Response · Authors · 2023-11-20
> **Dear Reviewer VLPY, [1]**
>
> Dear Reviewer VLPY,
>
> We appreciate your insightful comments.
>
> ---------------
>
> $\textbf{[Q1. Effect of rank on accuracy seems minimal]}$
>
> As astutely pointed out by the reviewer, the left side of Figure 5(a) might create the impression that the accuracy of LRQ (depicted by the red curve) remains relatively unchanged despite variations in the rank $r$, ranging from $64$ to $2048$. This could potentially lead readers to misconstrue the significance of the rank $r$ in influencing accuracy. To mitigate the risk of such misinterpretation, we conduct a comprehensive evaluation of the zero-shot and five-shot accuracies of LRQ for Llama 7B on common sense reasoning (CSR) and MMLU, considering the rank $r$ spanning from $64$ to $8192$.
>
> <Table A. Zero-shot and five-shot accuracies of LRQ for Llama 7B on common sense reasoning (CSR) and MMLU according to the rank $r$>
>
> | rank $r$ | 64 | 128 | 256 | 512 | 1024 | 2048 | 4096 | 8192 |
> |:------------|:---:|:---:|:---:|:---:|:---:|:---:|:---:|:---:|
> | Zero-shot accuracy of LRQ on CSR | 60.68 | 60.72 | 60.75 | 60.46 | 60.71 | 60.55 | 60.38 | 60.49 |
> | Five-shot accuracy of LRQ on MMLU | 33.97 | 33.90 | 34.25 | 34.20 | 34.47 | 34.07 | 33.21 | 31.51 |
>
> As outlined in Figure 9 in Appendix G of the revised manuscript, the accuracy of LRQ either remains relatively stable (Figure 9(a)) or increases gradually (Figure 9(b)) with the rise in the rank $r$ from $64$ to $1024$. However, as the rank $r$ continuously increases from $2048$ to $8192$, the accuracy of LRQ eventually declines to match that of FlexRound (indicated by the blue curve) on both CSR and MMLU. In light of these findings, it can be concluded that the employment of lower rank contributes considerably to improving the accuracy/generalization of quantized LLMs.
>
> ---------------
>
> $\textbf{[Q2. Using the scale of 0.8 for SmoothQuant is unsure]}$
>
> Although it is challenging to definitively assert that using $\alpha=0.8$ for SmoothQuant ensures its best performance, we adopted $\alpha=0.8$ on the grounds that the authors of SmoothQuant utilized $\alpha=0.8$ for all Llama models. Even when employing $\alpha=0.5$, a value considered by the authors of SmoothQuant as a general sweet spot, SmoothQuant ($\alpha=0.5$) typically exhibits lower performance compared to SmoothQuant ($\alpha=0.8$), as depicted in the tables below. We therefore believe that opting for $\alpha=0.8$ is not a suboptimal choice for SmoothQuant.
>
> <Table B. Zero-shot performance of SmoothQuant ($\alpha=0.5$) and SmoothQuant ($\alpha=0.8$) on common sense reasoning tasks>
>
> |  | W/A/KV | Llama 7B | Llama 13 B | Llama 33 B | Llama 65 B |
> |:----------------|:---:|:---:|:---:|:---:|:---:|
> | SmoothQuant ($\alpha=0.5$) | 8/8/16 | 56.93 | $\mathbf{60.10}$ | 60.13 | 64.85 |
> | SmoothQuant ($\alpha=0.8$) | 8/8/16 | $\mathbf{57.48}$ | 60.00 | $\mathbf{60.71}$ | $\mathbf{65.05}$ |
> | SmoothQuant ($\alpha=0.5$) | 8/8/8 | 57.19 | 59.30 | $\mathbf{60.77}$ | 65.02 |
> | SmoothQuant ($\alpha=0.8$) | 8/8/8 | $\mathbf{57.25}$ | $\mathbf{59.65}$ | $60.69$ | $\mathbf{65.24}$ |
>
> <Table C. Five-shot accuracy of SmoothQuant ($\alpha=0.5$) and SmoothQuant ($\alpha=0.8$) on MMLU>
>
> |  | W/A/KV | Llama 7B | Llama 13 B | Llama 33 B | Llama 65 B |
> |:----------------|:---:|:---:|:---:|:---:|:---:|
> | SmoothQuant ($\alpha=0.5$) | 8/8/16 | 29.53 | 29.43 | 41.97 | 52.93 |
> | SmoothQuant ($\alpha=0.8$) | 8/8/16 | $\mathbf{29.61}$ | $\mathbf{29.73}$ | $\mathbf{44.82}$ | $\mathbf{53.79}$ |
> | SmoothQuant ($\alpha=0.5$) | 8/8/8 | 29.18 | 29.27 | 42.29 | 52.63 |
> | SmoothQuant ($\alpha=0.8$) | 8/8/8 | $\mathbf{29.94}$ | $\mathbf{29.87}$ | $\mathbf{45.07}$ | $\mathbf{53.72}$ |
>
> -------------
>
> $\textbf{[Q3. Comparison of LRQ with SmoothQuant seems unfair]}$
>
> As noted by the reviewer, SmoothQuant is a training-free weight-activation post-training quantization (PTQ) method. However, the authors of SmoothQuant showcased its state-of-the-art performance among existing LLM weight-activation PTQ methods encompassing both learning-free and learning-based approaches. Consequently, we undertake a comparative analysis between our method, LRQ, and SmoothQuant, acknowledging the latter as one of the current state-of-the-art PTQ methods for LLM weight-activation quantization.
>
> To introduce another learning-based LLM weight-activation quantization technique, we compare LRQ with LLM-QAT, one of the state-of-the-art LLM weight-activation quantization-aware training (QAT) methods, for Llama 7B, 13B, and 33B on MMLU. As shown in the table below, LRQ demonstrates a substantial performance advantage over LLM-QAT, which reinforces the efficacy of LRQ further.
>
> <Table D. Five-shot accuracy of LLM-QAT and LRQ on MMLU>
>
> |  | W/A/KV | Llama 7B | Llama 13 B | Llama 33B |
> |:----------------|:---:|:---:|:---:|:---:|
> | LLM-QAT | 4/8/8 | 30.00 | 41.20 | 54.40 |
> | LRQ | 4/8/8 | $\mathbf{35.04}$ | $\mathbf{45.31}$ | $\mathbf{57.72}$ |
>
> -------------

---

> > ### Author Response · Authors · 2023-11-20
> > **Dear Reviewer VLPY, [2]**
> >
> > $\textbf{[Q4. Small difference between FlexRound and LRQ in Table 1]}$
> >
> > As the reviewer mentioned, in Table 1, LRQ exhibits a slightly superior zero-shot performance on common sense reasoning (CSR) compared to FlexRound. However, we believe that this advantage is noteworthy since FlexRound already achieves the zero-shot performance on CSR comparable to FP16 baselines. The close proximity in zero-shot performance between FlexRound and FP16 baselines on CSR limits the potential for a substantial performance disparity between FlexRound and LRQ. Despite LRQ approaching the zero-shot performance of FP16 baselines more closely than FlexRound, the difference in zero-shot performance between FlexRound and LRQ cannot be anticipated to be large after all.
> >
> > Nonetheless, as expounded in Section 1, it is crucial to emphasize that LRQ demonstrates competitive performance relative to FP16 baselines on both common sense reasoning (CSR) and MMLU, a feat not accomplished by FlexRound that excels solely on CSR. Given the comprehensive evaluation of large language models (LLMs) across diverse benchmarks, the proficiency of LRQ in excelling across both CSR and MMLU holds significant implications in the field of LLM quantization.
> >
> > Regarding the $8$-bit weight quantization presented in Tables 3 and 4, the adoption of a per-token asymmetric activation quantization scheme results in even naive rounding-to-nearest (RTN) performing closely to the levels of FP16 baselines on both CSR and MMLU. As a result, while LRQ exhibits slightly higher accuracy compared to SmoothQuant and FlexRound for most Llama 2 models, it can be concluded that SmoothQuant, FlexRound, and LRQ are nearly evenly matched.
> >
> > In the context of $4$-bit weight quantization as presented in Table 3, FlexRound achieves zero-shot accuracy levels comparable to FP16 baselines on CSR, resulting in a relatively small zero-shot performance gap between FlexRound and LRQ, like the scenario depicted in Table 1. However, in the case of $4$-bit weight quantization in Table 4, LRQ surpasses FlexRound by a margin ranging from $0.2$ to $0.7$ percent. Although, as noted by the reviewer, these increments in five-shot accuracy on MMLU in Table 4 may seem modest compared to those in Table 2, we believe that the rise in five-shot accuracy by $0.2$ to $0.7$ percent is significant. This is particularly noteworthy as it brings the five-shot accuracy gap between LRQ and FP16 baselines to less than $1.5$ percent, while the corresponding gap between FlexRound and FP16 baselines remains more or less at two percent for Llama 2 13B and 70B. We hope that these W4A8 quantization results will contribute to narrowing the gap between W4A8 quantized LLMs and FP16 baselines to less than one percent.
> >
> > -------------
> >
> > $\textbf{[Q5. Exponentiation on S in FlexRound]}$
> >
> > As the reviewer mentioned, there is no exponential notation for $\mathbf{S}\_2 \in \mathbb{R}\_{>0}^{C_{out} \times C_{in}}$ in the original FlexRound paper. However, to keep every entry of $\mathbf{S}\_2$ positive during minimizing the block reconstruction error, the authors of FlexRound express $\mathbf{S}\_2$ as $\text{exp}(\mathbf{S}\_2)$ where $\mathbf{S}\_2 \in \mathbb{R}^{C_{out} \times C_{in}}$  in the code implementation of FlexRound at the following url:
> > https://openreview.net/attachment?id=-tYCaP0phY_&name=supplementary_material. Accordingly, we also used a form of exponential notation in explaining FlexRound and LRQ.
> >
> > --------------

---

> > > ### Author Response · Authors · 2023-11-20
> > > **Dear Reviewer VLPY, [3]**
> > >
> > > $\textbf{[Q6. Ablation study on the effect of $\mathbf{r}\_2$ and $\mathbf{c}\_2$]}$
> > >
> > > Thank you for your helpful suggestion. We compare FlexRound, FlexRound with $\mathbf{S}\_2 = \mathbf{L}\_2 \mathbf{U}\_2$, and LRQ for Llama 7B and 13B.
> > >
> > > <Table E. Zero-shot performance of FlexRound, FlexRound with $\mathbf{S}_2 = \mathbf{L}_2 \mathbf{U}_2$, and LRQ on common sense reasoning tasks>
> > >
> > > |  | W/A/KV | Llama 7B | Llama 13 B |
> > > |:----------------|:---:|:---:|:---:|
> > > | FlexRound | 8/8/16 | 60.53 | 62.40 |
> > > | FlexRound with $\mathbf{S}\_2 = \mathbf{L}\_2 \mathbf{U}\_2$ | 8/8/16 | 60.69 | 62.62 |
> > > | LRQ | 8/8/16 | $\mathbf{60.71}$ | $\mathbf{62.92}$ |
> > > | FlexRound | 8/8/8 | 60.35 | 62.37 |
> > > | FlexRound with $\mathbf{S}\_2 = \mathbf{L}\_2 \mathbf{U}\_2$ | 8/8/8 | 60.49 | 62.62 |
> > > | LRQ | 8/8/8 | $\mathbf{60.63}$ | $\mathbf{62.76}$ |
> > >
> > > <Table F. Five-shot accuracy of FlexRound, FlexRound with $\mathbf{S}_2 = \mathbf{L}_2 \mathbf{U}_2$, LRQ on MMLU>
> > >
> > > |  | W/A/KV | Llama 7B | Llama 13 B |
> > > |:----------------|:---:|:---:|:---:|
> > > | FlexRound | 8/8/16 | 30.20 | 43.82 |
> > > | FlexRound with $\mathbf{S}\_2 = \mathbf{L}\_2 \mathbf{U}\_2$ | 8/8/16 | 33.86 | 45.48 |
> > > | LRQ | 8/8/16 | $\mathbf{34.47}$ | $\mathbf{45.83}$ |
> > > | FlexRound | 8/8/8 | 29.43 | 43.60 |
> > > | FlexRound with $\mathbf{S}\_2 = \mathbf{L}\_2 \mathbf{U}\_2$ | 8/8/8 | 33.96 | 45.21 |
> > > | LRQ | 8/8/8 | $\mathbf{34.39}$ | $\mathbf{45.83}$ |
> > >
> > > As evident from the tables above, FlexRound with $\mathbf{S}_2 = \mathbf{L}_2 \mathbf{U}_2$ surpasses the performance of FlexRound but falls short of LRQ. It is noteworthy that the five-shot accuracy on MMLU can witness an increase ranging from 1.5% to 4% by simply substituting $\mathbf{S}_2$ with $\mathbf{L}_2 \mathbf{U}_2$, which corroborates the significance of leveraging the parameter-efficiency inherent in low-rank weight-scaling matrices.
> > >
> > > -------------
> > >
> > > $\textbf{[Q7. Recommendation for the rank, learning rates, and training regimes]}$
> > >
> > > Drawing from our experience, a value of $r=2048$ would be appropriate for LLMs beyond 30B parameters, while $r=1024$ would be well-suited for smaller models. Regarding learning rates, in the context of a per-tensor static activation quantization scheme, a learning rate ranging between $1$e-$3$ and $3$e-$3$ is recommended. Conversely, when employing a per-token activation quantization scheme, a learning rate within the range of $1$e-$4$ to $5$e-$4$ is deemed suitable. The remainder of LRQ's experimental settings align with those detailed in [1].
> > >
> > > [1] Lee et al., FlexRound: Learnable Rounding based on Element-wise Division for Post-Training Quantization, ICML 2023.
> > >
> > > ------------
> > >
> > > $\textbf{[Q8. Presentation of implementation details in tabular form]}$
> > >
> > > Thank you for your practical suggestion. We present implementation details in tabular form in Appendix C of the revision.
> > >
> > > ------------
> > >
> > > $\textbf{[Q9. Typos]}$
> > >
> > > Thank you for pointing out typos. We fix those typos in the revised version.
> > >
> > > ------------

---

### Official Review · Reviewer_F2mg · 2023-10-31

**Soundness:** 2 fair
**Presentation:** 3 good
**Contribution:** 2 fair
**Rating:** 6
**Confidence:** 3

**Summary:**

This paper introduces LRQ, a post-training quantization technique for LLM (Large Language Models), which improves upon the FlexRound methods by reducing the size of weight-scaling matrices to enhance generalization performance. Through a comprehensive set of experiments conducted on Llama and Llama2, LRQ demonstrates improvements for 4-bit and 8-bit quantization in both Common Sense Reasoning (CSR) and Massive Multitask Language Understanding (MMLU) tasks.

**Strengths:**

1. The paper is well-written, and the low-rank weight-scaling method is simple and efficient, making it easy to comprehend.
2. This paper has conducted an extensive ablation study on the size of parameters and calibration data, providing valuable empirical insights into the optimal cost of the weight-scaling method.

**Weaknesses:**

1. LRQ builds upon FlexRound by introducing a low-rank matrix into the full weight-scaling method. As this is my first time as an ICLR reviewer, I am uncertain whether the novelty meets ICLR's standards.
2. This paper extensively explores experiments on both CSR and MMLU tasks, using various quantization methods. However, it consistently shows only marginal improvements compared to FlexRound, with one exception being MMLU when employing per-tensor asymmetric static activation quantization. Yet, the performance gain with per-token asymmetric activation quantization, which offers finer granularity, remains limited.

**Questions:**

1. It appears that LRQ applies low-rank quantization to weights. When it comes to activation quantization, is RTN used?

---

> ### Author Response · Authors · 2023-11-20
> **Dear Reviewer F2mg,**
>
> Dear Reviewer F2mg,
>
> We appreciate your constructive comments.
>
> ---------------
>
> $\textbf{[Q1. Novelty of LRQ]}$
>
> The use of low-rank matrices in LRQ seems to be simple and straightforward, yet we believe that it is important to underscore the underlying motivation of LRQ as the reviewer ok57 mentioned. As SmoothQuant scales the weights collectively per channel, SmoothQuant may lead to non-negligible accuracy loss after quantization. Since FlexRound learns an individual scale for every weight with limited calibration samples, FlexRound might be prone to overfitting. However, increasing calibration samples in FlexRound is not a viable option, which leads us to employ low-rank weight-scaling matrices in order to decrease the number of learnable parameters effectively while maintaining the concept of scaling weights individually by sharing learnable parameters via low-rank structure. In this regard, the novelty of our paper lies not merely in the utilization of low-rank matrices but is firmly rooted in the problem statement concerning the number of scaling parameters and a middle-ground approach to solving this challenge.
>
> ---------------
>
> $\textbf{[Q2. LRQ shows marginal improvements compared to FlexRound, with one exception of MMLU when using per-tensor asymmetric static activation quantization.]}$
>
> In Table 1, LRQ exhibits a slightly superior zero-shot performance on common sense reasoning (CSR) compared to FlexRound. However, we believe that this advantage is noteworthy since FlexRound already achieves the zero-shot performance on CSR comparable to FP16 baselines. The close proximity in zero-shot performance between FlexRound and FP16 baselines on CSR limits the potential for a substantial performance disparity between FlexRound and LRQ. Despite LRQ approaching the zero-shot performance of FP16 baselines more closely than FlexRound, the difference in zero-shot performance between FlexRound and LRQ cannot be anticipated to be large after all.
>
> Nonetheless, as expounded in Section 1, it is crucial to emphasize that LRQ demonstrates competitive performance relative to FP16 baselines on both common sense reasoning (CSR) and MMLU, a feat not accomplished by FlexRound that excels solely on CSR. Given the comprehensive evaluation of large language models (LLMs) across diverse benchmarks, the proficiency of LRQ in excelling across both CSR and MMLU holds significant implications in the field of LLM quantization.
>
> Regarding the $8$-bit weight quantization presented in Tables 3 and 4, the adoption of a per-token asymmetric activation quantization scheme results in even naive rounding-to-nearest (RTN) performing closely to the levels of FP16 baselines on both CSR and MMLU. As a result, while LRQ exhibits slightly higher accuracy compared to SmoothQuant and FlexRound for most Llama 2 models, it can be concluded that SmoothQuant, FlexRound, and LRQ are nearly evenly matched.
>
> In the context of $4$-bit weight quantization as presented in Table 3, FlexRound achieves zero-shot accuracy levels comparable to FP16 baselines on CSR, resulting in a relatively small zero-shot performance gap between FlexRound and LRQ, like the scenario depicted in Table 1. However, in the case of $4$-bit weight quantization in Table 4, LRQ surpasses FlexRound by a margin ranging from $0.2$ to $0.7$ percent. Although, as noted by the reviewer, these increments in five-shot accuracy on MMLU in Table 4 may seem modest compared to those in Table 2, we believe that the rise in five-shot accuracy by $0.2$ to $0.7$ percent is significant. This is particularly noteworthy as it brings the five-shot accuracy gap between LRQ and FP16 baselines to less than $1.5$ percent, while the corresponding gap between FlexRound and FP16 baselines remains more or less at two percent for Llama 2 13B and 70B. We hope that these W4A8 quantization results will contribute to narrowing the gap between W4A8 quantized LLMs and FP16 baselines to less than one percent.
>
> ---------------
>
> $\textbf{[Q3. When it comes to activation quantization, is RTN used?]}$
>
> Yes, we employ rounding-to-nearest (RTN) for activation quantization.
>
> ---------------

---

### Official Review · Reviewer_m3qc · 2023-10-31

**Soundness:** 3 good
**Presentation:** 3 good
**Contribution:** 2 fair
**Rating:** 6
**Confidence:** 3

**Summary:**

This work propose LRQ (Low-Rank Quantization), a simple yet effective PTQ method that leverages low-rank weight scaling matrices. This enables the method to learn with fewer parameters while still able to scale weights individually. Results show that the proposed method achieved better performance than existing works on both INT8 and INT4/8 cases.

**Strengths:**

(1) A thorough, detailed, easy-to-follow description of the method in section 2.

(2) A thorough evaluation of the proposed and related methods on various models and tasks.

**Weaknesses:**

(1) It would be helpful to add a comparison of the computation cost of the proposed and related quantization methods.

(2) Overall this is a solid work, but some of the results are not that exciting. For example, the authors made a big claim about INT4/8 in the abstract “Remarkably, we confirm for the first time the possibility of 4-bit weight and 8-bit activation quantization for LLMs with minimal accuracy loss among LLM PTQ studies.”. However, results in table 3 and 4 show that existing methods can do INT4/8 as well, sometimes even better than the proposed method.

**Questions:**

Please see my concerns in weaknesses.

---

> ### Author Response · Authors · 2023-11-20
> **Dear Reviewer m3qc,**
>
> Dear Reviewer m3qc,
>
> We appreciate your constructive comments.
>
> ---------------
>
> $\textbf{[Q1. Comparison of computational cost]}$
>
> For a comparative analysis of FlexRound and LRQ in terms of computational cost, we measure the execution time and peak GPU memory usage while quantizing Llama 7B to W8A8 using $512$ calibration samples and a batch size of $2$. FlexRound required 5 hours and 7 minutes, whereas LRQ consumed 5 hours and 22 minutes. LRQ's extended processing time is attributed to the multiplication involving $\mathbf{L}_2$ and $\mathbf{U}_2$. Despite the slightly longer runtime, LRQ demonstrates an advantage in peak GPU memory usage, utilizing $23.5$ GB compared to FlexRound's $25.4$ GB. This efficiency is attributed to LRQ's fewer learnable parameters in comparison to FlexRound.
>
> ---------------
>
> $\textbf{[Q2. Experimental results of W4A8 quantization]}$
>
> Although LRQ always outperforms FlexRound in the case of W4A8 quantization, as highlighted by the reviewer, the W4A8 performance of FlexRound on common sense reasoning tasks and MMLU are not notably poor. However, our primary intent was to underscore that we conducted experiments specifically addressing W4A8 quantization for the first time. In the original FlexRound paper, there were no experimental results pertaining to W4A8 quantization. To enhance clarity, we revise that portion of our manuscript to state, "we for the first time conduct experiments on W4A8 quantization among LLM PTQ studies.".
>
> ---------------

---

### Official Review · Reviewer_ok57 · 2023-10-31

**Soundness:** 3 good
**Presentation:** 4 excellent
**Contribution:** 3 good
**Rating:** 6
**Confidence:** 4

**Summary:**

Thanks to the authors for a well written paper.

The paper proposes a weight-activation quantization method based on low-rank weight scaling matrices. It does a nice job contrasting 2 preceding works (smoothquant & flexround) via the number of scaling parameters they each have. The authors take a middle of the ground approach which they show to be empirically beneficial, and motivate it as well using a generalization argument.

**Strengths:**

- good job explaining background work in smoothquant and flex round, and how they differ via number of scaling parameters
- nice motivation from flex round, ie using fig2 to show that increasing calibration samples is not viable, so therefore decrease learnable parameters
- pretty thorough set of experimental results

**Weaknesses:**

- how meaningful is the improvement of LRQ over FlexRound? is it possible to share std dev for the metrics you reported? For example in Table 1 Llama2 70B, the difference between LRQ has an average accuracy of 65.93 vs FlexRound 65.86. I see that sometimes the gap between LRQ and FlexRound is bigger (table 1: Llama33B), but also sometimes FlexRound is better (table 1: llama2 7b). I see that the gaps are much more sizable in Table2 (MMLU), but then smaller again for Tables 3 and 4.
- going back to the improved generalization motivation, if this interpretation were correct then wouldn’t we expect to see better performance on the common sense reasoning and MMLU tasks? I agree that LRQ’s performance is slightly better, but based on my previous comment I’m not sure how significant the gap is.

**Questions:**

- How many calibration samples were used to generate Fig 3? Also how large was the held out set? I understand the generalization argument, I’d just like to know how sensitive (or not) it is to the number of samples.
- It took me a couple minutes to realize the difference between Tables 1 and 2 was the evaluation task (common sense reasoning vs mmlu). I was confused because “per-tensor” and “static” were italicized. Just a small point, to somehow make it easier to digest.
- could you clarify your comment that LRQ is the first to confirm the possibility or 4Wa8 quantization? Because in Tables 3,4 LRQ does not do that much better than FlexRound.

Overall, I think the motivation and presentation of the method are very nice. The proposed low-rank solution makes a lot of sense, though methodologically there is nothing novel in the low-rank formulation. My main concern is in having a more careful interpretation of the empirical results (If I missed any, please point me to the paper). To me it seems like the gap between LRQ and FlexRound may only be significant in Table 2, and somewhat small elsewhere. Also, the claim that this methods makes 4W8A possible for the first time confuses me, as FlexRound is not that much worse (Tables 3&4). Is this claim really true? Please let me know if I am misunderstanding something. My reading of the results is that LRQ non-trivially increases the performance on MMLU tasks when using the setting of Table 2, and has minor improvements in other cases.

---

> ### Author Response · Authors · 2023-11-20
> **Dear Reviewer ok57, [1]**
>
> Dear Reviewer ok57,
>
> We appreciate your insightful comments.
>
> ---------------
>
> $\textbf{[Q1. Standard deviation of FlexRound and LRQ]}$
>
> As the reviewer suggested, we carry out three random trials for Llama 2 7B, Llama 33B, and Llama 2 70B as outlined in Table 1, presenting the average and standard deviation of them.
>
> <Table A. Average and standard deviation of zero-shot performance of FlexRound and LRQ over three random trials on common sense reasoning tasks>
>
> |  | W/A/KV | Llama 2 7B | Llama 33 B | Llama 2 70B |
> |:----------------|:---:|:---:|:---:|:---:|
> | FlexRound | 8/8/8 | $59.72 \pm 0.73$ | $62.83 \pm 0.36$ | $65.65 \pm 0.30$ |
> | LRQ | 8/8/8 | $\mathbf{59.90 \pm 0.18}$ | $\mathbf{63.81 \pm 0.16}$ | $\mathbf{65.89 \pm 0.06}$ |
>
> As seen in the table above, not only does the average of LRQ surpass that of FlexRound, but the standard deviation of LRQ is also smaller than that of FlexRound, which strengthens our assertion that FlexRound might be prone to overfitting when applied to the quantization of LLMs.
>
> ---------------
>
> $\textbf{[Q2. Motivation about the improved generalization]}$
>
> As the reviewer mentioned, LRQ exhibits a slightly superior zero-shot performance on common sense reasoning (CSR) compared to FlexRound. However, we believe that this advantage is noteworthy since FlexRound already achieves the zero-shot performance on CSR comparable to FP16 baselines. The close proximity in zero-shot performance between FlexRound and FP16 baselines on CSR limits the potential for a substantial performance disparity between FlexRound and LRQ. Despite LRQ approaching the zero-shot performance of FP16 baselines more closely than FlexRound, the difference in zero-shot performance between FlexRound and LRQ cannot be anticipated to be large after all.
>
> Nonetheless, we posit two notable points. First of all, as expounded in Section 1, LRQ demonstrates competitive performance relative to FP16 baselines on both common sense reasoning (CSR) and MMLU, a feat not accomplished by FlexRound that excels solely on CSR. Given the comprehensive evaluation of large language models (LLMs) across diverse benchmarks, the proficiency of LRQ in excelling across both CSR and MMLU holds significant implications in the field of LLM quantization. Moreover, notwithstanding LRQ's marginally superior zero-shot accuracy compared to FlexRound, the smaller variance of LRQ, as evidenced in Table A in the previous question, suggests that leveraging the parameter-efficiency of low-rank weight-scaling matrices can help variance reduction. Consequently, it can be inferred that LRQ holds the potential for superior generalization compared to FlexRound.
>
> ---------------
>
> $\textbf{[Q3. Number of calibration samples used to generate Figure 3]}$
>
> We employed $512$ calibration samples from the C4 dataset to perform the quantization of Llama 7B to W8A8, consistent with the experimental setup detailed in Section 3. However, in Figure 3, we depicted (a) a single sample chosen from the pool of $512$ calibration samples, and (b) an individual unseen sample from both common sense reasoning and MMLU. For visual representation across various samples, three figures were incorporated in Appendix E of the revision, each depicting the accumulated RMSE for three distinct samples.
>
> ---------------
>
> $\textbf{[Q4. Sensitivity of Figure 3 to the number of calibration samples]}$
>
> Thank you for your helpful suggestion. We incorporate a figure illustrating the sensitivity of accumulated RMSE to the number of calibration samples in Appendix F of the revised version. Figure 8 in the revision displays the comparison of accumulated RMSE for FlexRound and LRQ at the last Transformer block of Llama $7$B with the number of calibration samples varying from $64$ to $512$. As depicted in Figure 8(a), the accumulated RMSE of the last Transformer block on a calibration sample diminishes with a reduction in the number of calibration samples. This phenomenon is because FlexRound and LRQ are more likely to be fitted to calibration samples as the number of calibration samples becomes smaller. Conversely, Figure 8(b) reveals that the accumulated RMSE of the last Transformer block on each unseen sample from common sense reasoning and MMLU decreases with a larger number of calibration samples.
>
> Notably, the pattern elucidated in Section 2.4 persists consistently across varying calibration sample sizes from $64$ to $512$. In other words, for every calibration sample size spanning from $64$ to $512$, LRQ consistently attains nearly identical accumulated RMSE to FlexRound for a calibration sample from the C4 dataset. Concurrently, the accumulated RMSE of LRQ remains markedly smaller than that of FlexRound for an unseen sample from common sense reasoning and MMLU. This observation provides additional support for the insight presented in Figure 3, as discussed in Section 2.4.
>
> ---------------

---

> ### Author Response · Authors · 2023-11-20
> **Dear Reviewer ok57, [2]**
>
> $\textbf{[Q5. Italicization of “per-tensor” and “static” is confusing]}$
>
> Our initial aim was to clarify that Tables 1 and 2 present the experimental results of a per-tensor asymmetric static activation quantization scheme, while Tables 3 and 4 depict those of a per-token asymmetric activation quantization scheme. In the revised version, we refrain from italicizing "per-tensor" and "static" to prevent any potential confusion.
>
> ---------------
>
> $\textbf{[Q6. LRQ is the first to confirm the possibility of W4A8 quantization]}$
>
> As highlighted by the reviewer, the W4A8 performance of FlexRound on common sense reasoning tasks and MMLU are not notably poor. However, our primary intent was to underscore that we conducted experiments specifically addressing W4A8 quantization for the first time. In the original FlexRound paper, there were no experimental results pertaining to W4A8 quantization. To enhance clarity, we revise that portion of our manuscript to state, "we for the first time conduct experiments on W4A8 quantization among LLM PTQ studies.".
>
> ---------------
>
> $\textbf{[Q7. Novelty of low-rank formulation]}$
>
> The use of low-rank matrices in LRQ seems to be simple and straightforward, yet we believe that it is important to underscore the underlying motivation of LRQ as the reviewer mentioned. As SmoothQuant scales the weights collectively per channel, SmoothQuant may lead to non-negligible accuracy loss after quantization. Since FlexRound learns an individual scale for every weight with limited calibration samples, FlexRound might be prone to overfitting. However, increasing calibration samples in FlexRound is not a viable option, which leads us to employ low-rank weight-scaling matrices in order to decrease the number of learnable parameters effectively while maintaining the concept of scaling weights individually by sharing learnable parameters via low-rank structure. In this regard, the novelty of our paper lies not merely in the utilization of low-rank matrices but is firmly rooted in the problem statement concerning the number of scaling parameters and a middle-ground approach to solving this challenge.
>
> ---------------
>
> $\textbf{[Q8. Gap between LRQ and FlexRound may only be significant in Table 2 and somewhat small elsewhere]}$
>
> In Table 1, LRQ exhibits a slightly superior zero-shot performance on common sense reasoning (CSR) compared to FlexRound, which we believe is noteworthy since FlexRound already achieves the zero-shot performance on CSR comparable to FP16 baselines. The close proximity in zero-shot performance between FlexRound and FP16 baselines on CSR limits the potential for a substantial performance disparity between FlexRound and LRQ. Despite LRQ approaching the zero-shot performance of FP16 baselines more closely than FlexRound, the difference in zero-shot performance between FlexRound and LRQ cannot be anticipated to be large after all.
>
> Nevertheless, as expounded in Section 1, it is crucial to emphasize that LRQ demonstrates competitive performance relative to FP16 baselines on both common sense reasoning (CSR) and MMLU, a feat not accomplished by FlexRound that excels solely on CSR. Given the comprehensive evaluation of large language models (LLMs) across diverse benchmarks, the proficiency of LRQ in excelling across both CSR and MMLU holds significant implications in the field of LLM quantization.
>
> Regarding the $8$-bit weight quantization presented in Tables 3 and 4, the adoption of a per-token asymmetric activation quantization scheme results in even naive rounding-to-nearest (RTN) performing closely to the levels of FP16 baselines on both CSR and MMLU. As a result, while LRQ exhibits slightly higher accuracy compared to SmoothQuant and FlexRound for most Llama 2 models, it can be concluded that SmoothQuant, FlexRound, and LRQ are nearly evenly matched.
>
> In the context of $4$-bit weight quantization as presented in Table 3, FlexRound achieves zero-shot accuracy levels comparable to FP16 baselines on CSR, resulting in a relatively small zero-shot performance gap between FlexRound and LRQ, like the scenario depicted in Table 1. However, in the case of $4$-bit weight quantization in Table 4, LRQ surpasses FlexRound by a margin ranging from $0.2$ to $0.7$ percent. Although, as noted by the reviewer, these increments in five-shot accuracy on MMLU in Table 4 may seem modest compared to those in Table 2, we believe that the rise in five-shot accuracy by $0.2$ to $0.7$ percent is significant. This is particularly noteworthy as it brings the five-shot accuracy gap between LRQ and FP16 baselines to less than $1.5$ percent, while the corresponding gap between FlexRound and FP16 baselines remains more or less at two percent for Llama 2 13B and 70B. We hope that these W4A8 quantization results will contribute to narrowing the gap between W4A8 quantized LLMs and FP16 baselines to less than one percent.
>
> ---------------

---

### Author Response · Authors · 2023-11-23
**Dear Reviewers,**

Dear Reviewers,

We sincerely appreciate your time and efforts in reviewing our paper. We kindly remind you of our responses to your insightful and constructive comments and faithfully reflected them in the revised manuscript. As the discussion period will end in about 9 hours, we would appreciate if you check our responses and the revision. If you have any remaining issues or concerns, please do not hesitate to bring them to our attention.

Warm regards,

Authors

---

### Meta-Review · Area_Chair_dC2Z · 2023-12-10

**Metareview:**

Three reviewers gave a score of 6 but with fairly low confidence and little enthusiasm. One reviewer gave a score of 3 with high confidence and a more detailed review. All reviewers noted a limited methodological novelty, marginal improvements over previous work, and overall the results are not that exciting. Unfortunately, the area of neural net compression has been very crowded for years and it is hard to make an impression. Unfortunately, this paper doesn't have enough support for acceptance.

**Justification For Why Not Higher Score:**

See metareview

**Justification For Why Not Lower Score:**

N/A

---

### Decision · Program_Chairs · 2024-01-16

Reject